# Comparative analysis of multiple DNA double-strand break repair pathways in CRISPR-mediated endogenous tagging

Chiharu Tei[1,5], Shoji Hata [1,2,5 ✉], Akira Mabuchi[1,5], Shotaro Okuda[1], Kei K. Ito[1], Mariya Genova [3], Masamitsu Fukuyama[1], Shohei Yamamoto[1], Takumi Chinen[1], Atsushi Toyoda [4] & Daiju Kitagawa [1 ✉]

CRISPR-mediated endogenous tagging is a powerful tool in biological research. Inhibiting the non-homologous end joining (NHEJ) pathway has been shown to improve the low efficiency of accurate knock-in via homology-directed repair (HDR). However, the influence of alternative double-stranded break (DSB) repair pathways on knock-in remains to be fully explored. In this study, our long-read amplicon sequencing analysis reveals various patterns of imprecise repair in CRISPR-mediated knock-in, even with NHEJ inhibition. Further suppressing either microhomology-mediated end joining (MMEJ) or single-strand annealing (SSA) reduces nucleotide deletions around the cut site, thereby elevating knock-in accuracy. Additionally, imprecise donor integration is reduced by inhibiting SSA, but not MMEJ. Particularly, SSA suppression reduced asymmetric HDR, a specific imprecise integration pattern, which we further confirm using a novel reporter system. These findings demonstrate the complex interplay of multiple DSB repair pathways in CRISPR-mediated knock-in and offer novel strategies, including SSA pathway targeting, to improve precise gene editing efficiency.

The CRISPR/Cas-mediated endogenous gene tagging system is an important tool for analyzing protein localization and function in their native context by introducing a DNA sequence encoding for a peptide or a protein tag into the gene of interest. In this system, Cas endonucleases are targeted to the specific gene locus with a guide RNA to induce double-strand DNA breaks (DSBs). Cpf1 (Cas12a) and Cas9, which are the most commonly used Cas nucleases, recognize different PAM sequences and cleave DNA in distinct manners[1–3]. These DSBs can be repaired through the homology-directed repair (HDR) pathway, which uses a homologous DNA sequence as a template to seal the DSB precisely. Exogenously introduced DNA, containing so-called homology arms (HAs) — elements that are homologous to the regions flanking the DSB site, can also serve as a repair template. Therefore, a desired sequence flanked by HAs in such a donor DNA template can be accurately integrated into a specific gene locus via the HDR pathway in CRISPR/Cas-mediated gene knock-in[4].

Despite the broad applications of this method, the low efficiency of accurate editing events remains a significant challenge[5]. This issue arises from various DSB repair outcomes other than the precise incorporation of the donor DNA sequences, which is referred to as "perfect HDR"[6]. For example, DSBs can be repaired without utilizing donor DNA, leading either to a seamless reconstitution of the wild-type (WT) sequence or introducing insertions/deletions (indels) at the lesion locus. Moreover, even when the DSB repair is dependent on the donor DNA, imprecise donor integration into the target site frequently occurs. The imprecise incorporation can result in various faulty repair patterns, such as partial integration of the donor sequence, HAs duplication, and "asymmetric HDR", where only one side of donor DNA is precisely integrated but the other is not[6]. The non-homologous end joining (NHEJ), which is recognized as the most dominant DSB repair pathway, ligates the free DNA ends at the lesion in a homology-independent manner at the cost of higher indel incidence[7]. Suppressing this pathway leads to a reduction of indel occurrence, thereby enhancing the efficiency of perfect HDR[8,9]. Based on this, inhibiting non-HDR repair pathways has been proposed as an effective strategy to increase the efficiency of precise knock-in[10].

In addition to NHEJ and HDR, there are two alternative non-HDR DSB repair pathways: microhomology-mediated end joining (MMEJ), and

[1]Department of Physiological Chemistry, Graduate School of Pharmaceutical Sciences, The University of Tokyo, Bunkyo, Tokyo, Japan. [2]Precursory Research for Embryonic Science and Technology (PRESTO) Program, Japan Science and Technology Agency, Honcho Kawaguchi, Saitama, Japan. [3]Zentrum für Molekulare Biologie, Universität Heidelberg, DKFZ-ZMBH Allianz, Heidelberg, Germany. [4]Comparative Genomics Laboratory and Advanced Genomics Center, National Institute of Genetics, Mishima, Shizuoka, Japan. [5]These authors contributed equally: Chiharu Tei, Shoji Hata, Akira Mabuchi. ✉e-mail: s.hata@mol.f.u-tokyo.ac.jp; dkitagawa@mol.f.u-tokyo.ac.jp

https://doi.org/10.1038/s42003-025-08187-5                                                                                    **Article**

single-strand annealing (SSA)[11,12]. MMEJ relies on the annealing of two microhomologous sequences (2–20 nt) flanking the broken junction, which frequently results in introduction of deletions at the junction[13,14]. While the MMEJ pathway has been regarded as a minor DSB repair pathway compared to NHEJ, there is growing evidence that suppression of the MMEJ pathway by inhibiting its central effector POLQ increases HDR frequency in CRISPR-mediated knock-in[15–18]. In contrast, the last DSB repair pathway, the SSA pathway, utilizes Rad52-dependent annealing of longer homologous sequences for DSB repair[19]. In the context of CRISPR-mediated gene editing, SSA-mediated repair has been reported to occur in artificial gene cassettes containing two homologous regions, resulting in deletions of the intervening sequence between them[20,21]. Although these two non-HDR pathways have been suggested to take part in repairing DSBs introduced by Cas nucleases as mentioned, their impact on the repair outcomes in CRISPR/Cas-mediated knock-in remains unclear.

In this study, we investigated the contributions of these non-HDR pathways to the repair outcomes of CRISPR/Cas-mediated endogenous gene tagging with fluorescent proteins in human non-transformed diploid cells. Our observation revealed the presence of various imprecise repair patterns along with perfect HDR, even when NHEJ is inhibited. The inhibition of POLQ and Rad52, key components of the MMEJ and SSA pathways, respectively, resulted in a decrease in deletions around the target site, with SSA suppression showing effects dependent on the nature of DNA cleavage ends. Furthermore, we found that suppression of the SSA pathway resulted in decreased the occurrence of various donor mis-integration events, especially asymmetric HDR. Taken together, our findings indicate that the MMEJ and SSA pathways contribute to distinct patterns of imprecise editing, and thus their inhibition can further enhance the efficiency of precise gene knock-in in combination with NHEJ suppression.

## Results

### NHEJ inhibition is not sufficient to completely suppress non-HDR repairs in Cpf1- and Cas9- mediated endogenous tagging

We first examined the effects of inhibiting the three non-HDR pathways on knock-in outcomes in the hTERT-immortalized RPE1 cell line, which is a human non-transformed diploid cell line commonly used in the field of cell biology. For knock-in in RPE1 cells, we applied a cloning-free endogenous tagging method established previously[22]. Following this protocol, we prepared donor DNA by PCR using a pair of primers containing 90 bases of HA sequences. Recombinant Cas nucleases and guide RNAs transcribed in vitro were mixed to form RNP complexes, which were electroporated into cells along with the donor DNA (Fig. 1a). To examine the effects of inhibiting the three non-HDR repair pathways on both Cpf1- and Cas9-mediated knock-in, we performed a Cpf1-mediated C-terminal tagging of the nuclear protein HNRNPA1 and a Cas9-mediated N-terminal tagging of the trafficking protein RAB11A with the green fluorescent protein mNeonGreen (mNG). Immediately after the electroporation of either Cpf1-RNP or Cas9-RNP along with the donor DNA, these cells were treated for 24 h with specific inhibitors targeting each repair pathway (Fig. 1a). In line with most previous studies[15,23], we set the inhibitor treatment duration to 24 h, considering that HDR typically occurs within this timeframe after Cas9 protein delivery[24]. For NHEJ suppression, we employed Alt-R HDR Enhancer V2, a potent NHEJ inhibitor (NHEJi)[23]. Suppression of the MMEJ pathway was achieved using ART558, a recently discovered inhibitor of POLQ, which is the key enzyme driving this repair process[25]. For the inhibition of the SSA pathway, we utilized D-I03, a specific inhibitor targeting Rad52, which mediates the annealing of homologous single-stranded DNA (ssDNA) sequences[26,27]. Fluorescence imaging showed that most of the mNG-positive cells exhibited the expected localization corresponding to each mNG-fused endogenous protein (Fig. 1b, Figure S1). Consistent with previous reports[9,28], NHEJi treatment seemed to result in an increase in the cell population with the mNG signal (Fig. 1b, Figure S1), while inhibiting MMEJ or SSA had no notable effect (Fig. 1b). For precise quantification of knock-in efficiency, we performed flow cytometric analysis 4 days after electroporation. This analysis revealed that NHEJi treatment increased knock-in efficiency by

approximately 3-fold for both Cpf1-mediated knock-in at the *HNRNPA1* locus (5.2% to 16.8%) and Cas9-mediated knock-in at the *RAB11A* locus (6.9% to 22.1%). In contrast, inhibiting either the MMEJ or SSA pathway had no significant effect on knock-in efficiency (Fig. 1c, d). These results indicate that NHEJ is the predominant non-HDR pathway affecting knock-in efficiency in CRISPR-mediated endogenous tagging.

Next, we comprehensively analyzed the resulting repair patterns following the Cas nuclease-induced DNA DSBs at the target loci under the inhibition of the three non-HDR DSB repair pathways. For this purpose, we conducted the long-read amplicon sequencing using PacBio for knock-in alleles followed by genotyping using a computational framework called *knock-knock*[6] (Fig. 1e). After the electroporation of Cpf1-RNP for mNG tagging at the *HNRNPA1* locus followed by treatment with the specific pathway inhibitors, the knock-in target sites were amplified by PCR from extracted genomic DNA (Fig. 1e). Subsequent to the sequencing process, each Hi-Fi read was categorized through the *knock-knock* classification process into one of the possible DSB repair outcomes, such as WT, indels, perfect HDR, or subtypes of imprecise integration. Consistent with the flow cytometry results, NHEJ inhibition drastically increased the frequency of perfect HDR events while significantly reducing small deletions (<50 nt) compared to control samples (Fig. 1f). MMEJ inhibition also significantly increased perfect HDR frequency, which coincided with reduction in large ($\geq$ 50 nt) deletions and complex indels, consistent with previous studies[15,17,29–32]. In contrast, SSA inhibition had no substantial effect on repair pattern distribution. Notably, even when inhibiting NHEJ, the predominant non-HDR DSB repair pathway, the proportion of perfect HDR events was still far below 100% among all integration events (Fig. 1g). This phenomenon was consistent across all tested loci: Cpf1-mediated knock-in at *HNRNPA1* and *TOMM20* loci, as well as Cas9-mediated knock-in at *RAB11A* and *CLTA* loci, where imprecise integration still accounted for nearly half of all integration events despite NHEJ inhibition. These findings suggest that the other non-HDR DSB repair pathways may contribute to such imprecise integration when NHEJ is inhibited.

Subsequently, we performed a more detailed analysis of the imprecise integrations under NHEJ inhibition. Based on the classification by *knock-knock*, we placed the imprecise integration events into the following categories: blunt (both ends of the donor DNA including the HAs are directly ligated to the cut site), asymmetric HDR (only one side of the donor DNA is precisely integrated in an HDR manner), imperfect (both ends of the donor are integrated through a non-HDR pathway, with at least one end being trimmed), concatenated (multiple insertions of donors), and complex (not classified into the other four mis-integration categories). Among the imprecise integrations detected in both Cpf1- and Cas9-based endogenous tagging approaches, asymmetric HDR and imperfect integrations were the major patterns, together with complex integrations (Figure S2a). Interestingly, compared to Cpf1, the Cas9-based genome editing tended to have a higher percentage of blunt integration events. This observation can be attributed to the unique DNA ends at the DSB generated by these two Cas nucleases: blunt ends by Cas9 and staggered ends by Cpf1[1–3].

Similarly, in donor independent repair, Cas9-mediated knock-ins predominantly resulted in wild-type (WT) repair, likely due to blunt ligation of broken DNA ends (Figure S2b). In contrast, deletions smaller than 50 nt were the most frequent outcome of Cpf1-mediated cleavage, emphasizing the distinct DNA end structures generated by the two nucleases (Figure S2b). However, some variations in the repair patterns among the different gene loci could be detected, even when the same Cas nuclease were used. For example, in Cas9-mediated knock-ins, genomic integration, wherein DNA sequences from outside of the target locus incorporated into the target site, was predominantly observed at the *RAB11A* locus but rarely at the *CLTA* locus (Fig. 1g). These variations suggest that repair patterns depend on both the genomic context and the types of the applied Cas nucleases. Collectively, these data indicate that although NHEJ is the predominant repair pathway, diverse other non-HDR repair mechanisms can occur at high frequency under NHEJ inhibition during CRISPR-mediated gene knock-in.

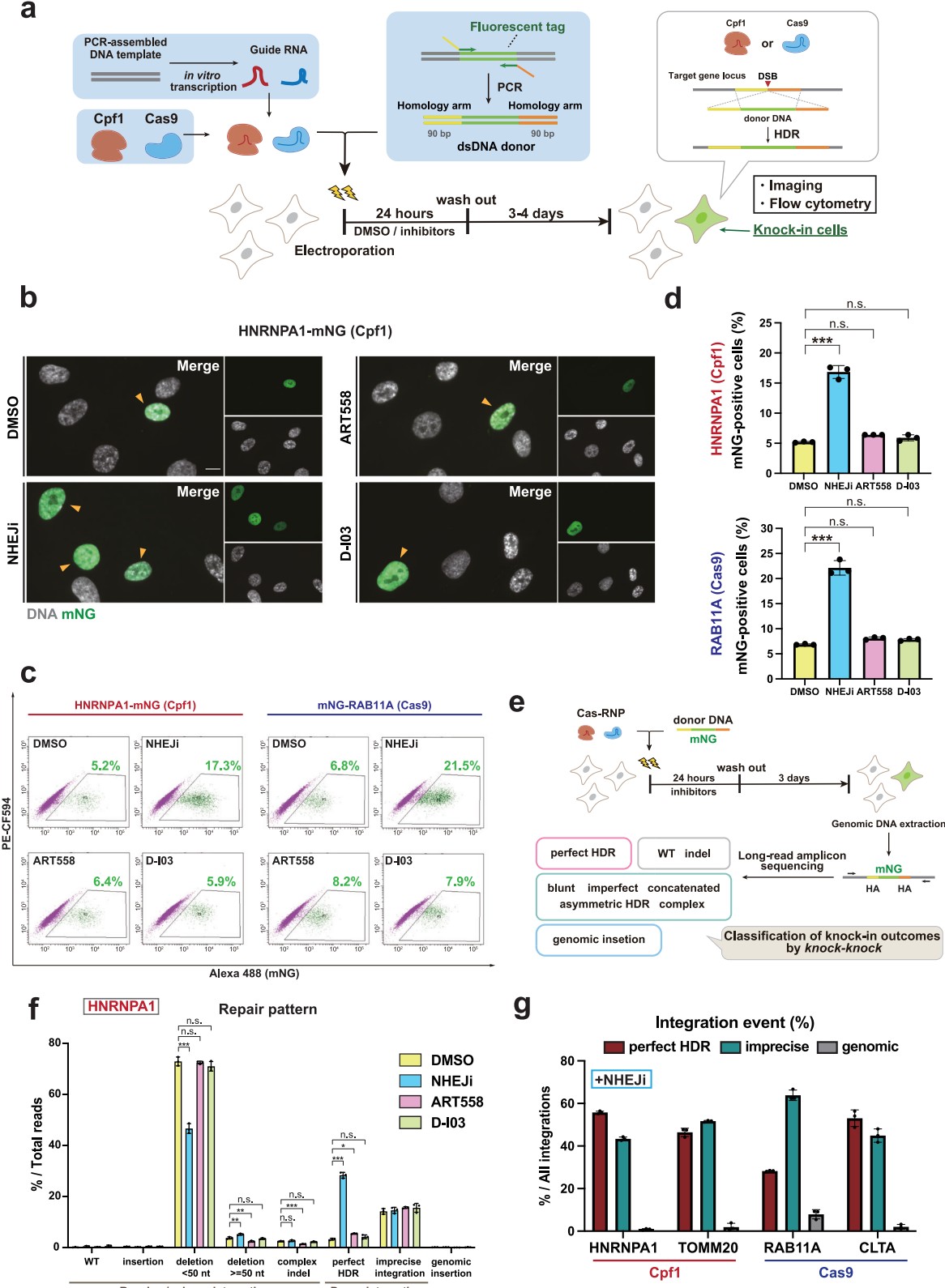

## The MMEJ and SSA non-HDR pathways influence the efficiency of gene knock-in upon NHEJ inhibition

Next, to examine the contribution of non-HDR pathways other than NHEJ to the knock-in process, we attempted to suppress the MMEJ and SSA pathways in addition to NHEJ inhibition. In order to detect changes in knock-in likelihood in a highly sensitive manner, we applied simultaneous

dual-color tagging with both mNG and the red fluorescent protein mScarlet, which can detect a substantial fraction of bi-allelic knock-in by analyzing mNG and mScarlet double-positive cells (Fig. 2a). The expected high sensitivity is based on a theoretical assumption that probability of bi-allelic editing increases proportionally to the square of editing probability[33,34]. Dual-color tagging of HNRNPA1 resulted in a comparable percentage of

**Fig. 1 | NHEJ inhibition is insufficient for complete suppression of imprecise repair. a** Schematic overview of endogenous gene tagging in human RPE1 cells. The cells were cultured in a medium containing DMSO or 1 µM NHEJi or 10 µM ART558 (POLQ inhibitor) or 10 µM D-I03 (Rad52 inhibitor) for 24 h after electroporation of Cas-RNP and the donor DNA. **b** Representative images from cells with Cpf1-mediated mNG tagging of HNRNPA1, treated with the indicated inhibitors for 24 h. Arrowheads indicate cells with nuclear mNG signals. Cells at 5 days after electroporation were fixed and analyzed. Scale bar: 10 µm. **c** Flow cytometric analysis of Cpf1-mediated HNRNPA1-mNG and Cas9-mediated mNG-RAB11A knock-in cells. Cells treated with the indicated inhibitors were analyzed 4 days after electroporation. Percentages of cells with mNG signal are shown in the plots. **d** Quantification of percentages of mNG-positive cells from **c**. Approximately 10,000

cells were analyzed for each sample. **e**, Schematic overview of sequencing-based approach to analyze overall repair patterns of CRISPR-mediated endogenous tagging. Amplicon sequencing using PacBio was performed on a knock-in target locus. Using the *knock-knock* algorithm, each sequence read was categorized into a specific category of knock-in outcomes. **f** Distribution of repair patterns classified by *knock-knock* under treatment of different inhibitors. 97,292-319,727 reads were analyzed for each sample. **g** Distribution of integration events (perfect HDR, imprecise donor integration, and genomic insertion) across the targeted genes. For each category, percentage within total integration events are shown. In **d**, **f**, **g**, data from three biological replicates are represented as mean ± S.D. and *P*-values were calculated by a Tukey–Kramer test in this figure. *$P < 0.05$, **$P < 0.01$, ***$P < 0.001$, n.s. Not significant.

---

total fluorescence-positive cells as single-color tagging of the same gene in both DMSO control and NHEJi conditions (Fig. 2b, c). As expected, inhibition of NHEJ had a much higher impact on the percentage of mNG and mScarlet double-positive cells (0.06% to 1.49%, more than 20 times higher) than on that of total fluorescence-positive cells (5.3% to 17.0%, about 3 times higher) in the dual-color tagging (Fig. 2b, c). These results indicate that detection of changes in bi-allelic knock-in efficiency by the dual-color tagging system provides a significantly more sensitive approach for evaluating changes in knock-in likelihood compared to measuring knock-in efficiency by the single-color tagging system.

To achieve simultaneous suppression of both NHEJ and MMEJ pathways, cells were co-treated with NHEJi and ART558 for 24 h. This dual inhibition resulted in an increased proportion of double-positive cells at the *HNRNPA1* locus compared to NHEJi treatment alone (Fig. 2d). We further validated the impact of MMEJ inhibition by siRNA-mediated knockdown of POLQ combined with NHEJi treatment (Fig. 2e, f). POLQ depletion also enhanced the rate of double-positive cells in Cpf1-mediated knock-in at the *HNRNPA3* and *TOMM20* loci, as well as in Cas9-mediated knock-in at the *RAB11A* locus (Fig. 2g), indicating a general effect of MMEJ suppression on improving knock-in efficiency under NHEJ inhibition. Similarly, suppression of the SSA pathway through D-I03 treatment or Rad52 depletion, combined with NHEJ inhibition, resulted in a significant increase in the proportion of double-positive cells for HNRNPA1 tagging (Fig. 2d–f). However, at the other loci, such effect was observed only at the *HNRNPA3* locus, but not at the *TOMM20* or *RAB11A* locus (Fig. 2g), suggesting that the impact of SSA pathway suppression on knock-in efficiency is locus-dependent. At the SSA-sensitive *HNRNPA1* locus, co-treatment with ART558 and D-I03 synergistically improved the efficiency of dual-color tagging under NHEJ inhibition (Fig. 2d). These findings reveal that these two minor DSB repair pathways MMEJ and SSA distinctly influence the CRISPR-mediated knock-in process when NHEJ is inhibited.

## MMEJ suppression enhances the frequency of perfect HDR by affecting donor-independent repair without influencing imprecise donor integration

We next assessed the influence of suppressing the minor non-HDR pathways MMEJ and SSA on the repair patterns of Cpf1-mediated mNG tagging of HNRNPA1 upon NHEJ inhibition. The long-read amplicon sequencing and subsequent *knock-knock* analysis revealed a noticeable increase in the proportion of perfect HDR within total reads when cells were treated with the MMEJ inhibitor ART558 under NHEJ inhibition (Fig. 3a). This increase can be attributed to a significant decrease in the proportion of deletions of less than 50 nt (Fig. 3a). Interestingly, MMEJ inhibition did not significantly affect imprecise donor integration (Fig. 3a), which challenges current understanding that MMEJ directly drives imprecise donor integration[16,35,36]. To further validate that MMEJ is not involved in imprecise donor integration, we analyzed the microhomology length distributions at genome-donor junctions in the imprecise integration events. The results showed that the combination of ART558 treatment with NHEJ inhibitor had no significant impact on either the total frequency of ≥2 bp microhomologies or the distribution of microhomology lengths (Fig. 3b, Figure S3c). These

findings were further corroborated in our Cpf1-mediataed mNG tagging experiments with TOMM20, where MMEJ inhibition similarly showed no impact on either total frequency of microhomology or microhomology length distribution (Figure S3a-c).

To investigate the specific features of reduced deletions by ART558, we extracted the sequencing reads that *knock-knock* classified as deletions of less than 50 nt and analyzed them using the Sequence Interrogation and Quantification (SIQ) program, which enabled us to analyze the positional distribution of deletions[37]. Intriguingly, in the condition where only NHEJ was inhibited, the frequency of deletions is higher in the upstream of the region flanked by the two cut sites than in the downstream of the region (Fig. 3c). This asymmetric deletion pattern was also observed in Cpf1-mediated knock-in at the *TOMM20* locus (Figure S3a, d). Since the cut site of Cpf1 in the strand targeted by crRNA is located outside of the target sequence, the DSB can be imprecisely repaired without disrupting target recognition by Cpf1, which is hypothesized to allow repeated cleavages until the deletion reaches within the target sequence[2]. This unique property of Cpf1 could result in a high frequency of deletions within the target sequence among various other patterns, leading to the biased distribution observed (Fig. 3c).

The SIQ analysis showed that, under conditions of NHEJ inhibition, ART558 decreased the frequency of deletions spanning a wide range around the cut site at the *HNRNPA1* locus (Fig. 3c). In particular, the deletions occurring in the downstream region of the cut site, flanked by two T-rich sequences (indicated in cyan), showed a pronounced decrease. This suggests that the MMEJ-mediated repair is efficiently facilitated by the multiple microhomologous areas available between the two T-rich sequences. Collectively, these findings indicate that while MMEJ plays a significant role in donor-independent repair processes that result in genomic deletions, it does not substantially contribute to imprecise donor integration during CRISPR-mediated endogenous tagging.

## SSA suppression enhances perfect HDR efficiency by reducing small genomic deletions and imprecise donor integration

Similar to MMEJ suppression, the inhibition of the SSA pathway through D-I03 treatment enhanced perfect HDR frequency by reducing deletions smaller than 50 nt at the *HNRNPA1* locus under NHEJ inhibition (Fig. 3a). Subsequent SIQ analysis on these deletion reads showed that D-I03 treatment selectively reduced deletions in the downstream region of the cut site, flanked by two T-rich sequences, while ART558 affected overall deletions around the cut site (Fig. 3c). Consistently, in Cpf1-mediated knock-in at the *TOMM20* locus, SSA inhibition selectively decreased the deletions within a region immediately downstream of the cut site (Figure S3d). These findings indicate that the SSA suppression enhances the efficiency of perfect HDR by reducing deletions through mechanisms distinct from MMEJ suppression.

Having revealed that MMEJ and SSA contribute to imprecise repair under NHEJ inhibition in distinct ways, we next investigated their combined inhibition effects in practical applications. In the process of establishing endogenously tagged cell lines with fluorescent proteins, flow cytometric cell sorting is routinely used to enrich successfully tagged cells. To evaluate the effects of the MMEJ or SSA pathway suppression in this context, we tagged

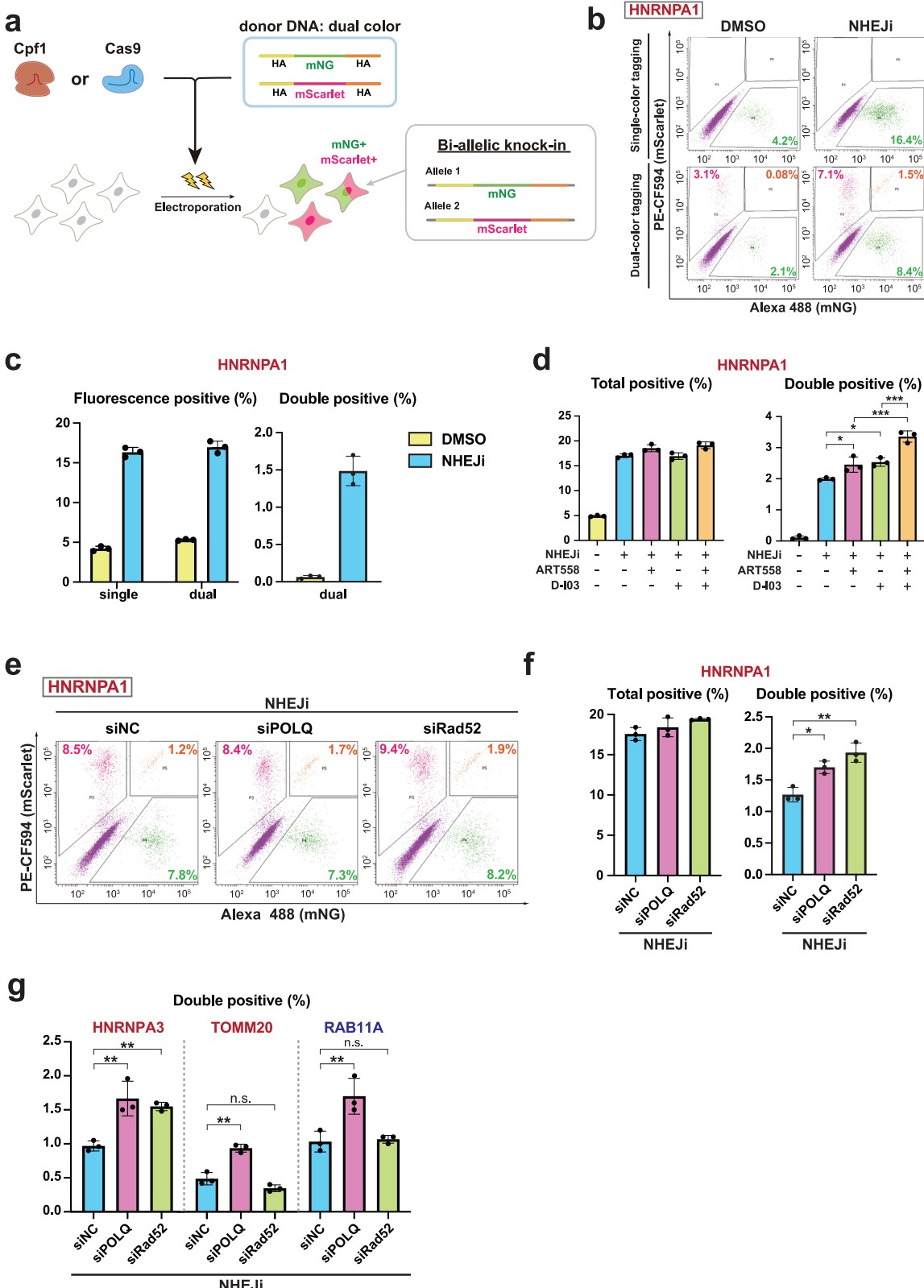

HNRNPA1 with mNG using Cpf1, collected 10,000 mNG-positive cells by flow cytometry, and conducted long-read amplicon sequencing and *knock-knock* analysis. The results showed that the overall proportions of the donor-independent repair outcomes, especially deletions of less than 50 nt, significantly decreased compared to those in non-sorted cells (Fig. 3a, d). Among sorted cells, MMEJ suppression under NHEJ inhibition had minimal effect on the repair patterns, unlike the non-sorted condition. In contrast, treatment with D-I03 in addition to NHEJi resulted in a significant increase in perfect HDR frequency even in the sorted population. Notably, this enhancement in perfect HDR stemmed from reduced imprecise donor integrations rather than the decrease in deletion as observed in non-sorted conditions.

**Fig. 2 | MMEJ or SSA suppression enhances endogenous tagging efficiency under NHEJ inhibition. a** Schematic of dual-color tagging, where two DNA donors containing mNG or mScarlet sequence were simultaneously introduced. **b** Flow cytometric analysis of cells with single-color tagging (mNG) and dual-color tagging (mNG and mScarlet) of HNRNPA1. After electroporation, cells were treated with DMSO or NHEJi for 24 h and subsequently cultured for 3 days before the analysis. The plots display the percentages of cells exhibiting only mNG signal (green), only mScarlet signal (magenta), or both (orange). **c** Quantification of percentages of fluorescence-positive (left) and double-positive cells (right) from **b**. In single-color tagging, fluorescence-positive cells refer to mNG-positive cells. As for dual-color tagging, fluorescence-positive cells correspond to those that are positive for either mNG or mScarlet, or both, while double-positive cells refer to those that exhibit signals for both mNG and mScarlet. Approximately 10,000 cells were analyzed for each sample. **d** Quantification of percentages of total- and double-positive cells in

dual-color tagging of HNRNPA1 using Cpf1. Cells were treated with the indicated inhibitors for 24 h after electroporation. More than 5000 cells were analyzed for each sample. **e** Flow cytometric analysis of dual-color tagged cells depleted of POLQ or Rad52 under NHEJ inhibition. siRNAs targeting the indicated genes were transfected 24 h before electroporation of Cas-RNP and donor DNA. After treatment with 1 μM NHEJi for 24 h and subsequent incubation in fresh media, cells were subjected to flow cytometric analysis. **f** Quantification of percentages of total- and double-positive cells in dual-color tagging from **e**. Approximately 10,000 cells were analyzed for each sample. **g** Quantification of percentages of double-positive cells for the indicated conditions in dual-color tagging of HNRNPA3 and TOMM20 using Cpf1, and RAB11A using Cas9. Approximately 10,000 cells were analyzed for each sample. In **c**, **d**, **f**, **g**, data from three biological replicates are presented as mean ± S.D. and *P* value was calculated by a Tukey–Kramer test in this figure. *P < 0.05, **P < 0.01, ***P < 0.001, n.s. Not significant.

To comprehensively analyze the impact of suppressing the SSA pathway on imprecise integration events, we analyzed the mis-integration patterns in both sorted and non-sorted conditions. For simplicity, we focused on three major categories of mis-integration events derived from only single-donor insertion, namely blunt integrations, asymmetric HDR, and imperfect integrations. In cells with NHEJ inhibition alone, sorted cells showed a higher percentage of asymmetric HDR events, where only one side of the donor DNA integrates precisely, compared to non-sorted ones. (Fig. 3e). This likely reflects the better preservation of functional fluorescent protein sequences in asymmetric HDR compared to other imprecise integration patterns. Notably, in sorted cells, SSA pathway inhibition with D-I03 tended to decrease the percentage of all three types of imprecise integrations. In particular, asymmetric HDR events, the most common type detected in sorted cells, exhibited a pronounced tendency for reduction. A similar reduction in asymmetric HDR events was further confirmed in the sorted cells through Cpf1-mediated mNG tagging of CLTC (Figure S4a, b). Imperfect integrations also exhibited a tendency to decrease upon SSA inhibition at the *HNRNPA1* and *CLTC* loci (Fig. 3e and Figure S4b). These findings indicate that, during endogenous tagging with fluorescent proteins, SSA pathway suppression enhances perfect HDR frequency in the fluorescence-positive cells primarily by reducing imprecise integrations, particularly asymmetric HDR and imperfect integration events. Collectively, our data reveals that SSA suppression results in enhancement of perfect HDR efficiency through distinct mechanisms depending on the experimental context.

### A flow cytometry-based reporter system further confirmed attenuation of asymmetric HDR upon SSA suppression

To further confirm whether the suppression of the SSA pathway effectively reduces asymmetric HDR, we engineered a flow cytometry-based assay, named "HDR reporter". In this reporter system, when each end of the donor is precisely inserted into the target site, cells emit distinct fluorescence wavelengths, which allows for simultaneous evaluation of both perfect and asymmetric HDR events (Fig. 4a). The HDR reporter cassette, containing the N-terminal half of mNG, a spacer sequence, and the C-terminal half of mScarlet, was inserted at the *HNRNPA1* locus in the genome of RPE1 cells. The spacer is flanked on both sides by identical Cas9 target sequences placed in the opposite directions, which originates from the mouse *Adenylate kinase 2* gene and exhibit minimal homology to the human genome. The cells were transfected with Cas9-RNP and the donor DNA, which contains the complementary C-terminal half of mNG, the 2A peptides sequence, and the complementary N-terminal half of mScarlet, all flanked by 100 bases of HA sequences. In this system, the seamless repair of the Cas9-induced DSBs in a perfect HDR manner will complete the two split fluorescent protein sequences, resulting in their simultaneous fluorescence emission. Conversely, when imprecise donor integration occur, indels are introduced in the middle of the fluorescent protein sequences. Since these introductions would very likely disrupt the functionality of the fluorescent protein, the HDR reporter is highly susceptible to imprecise integrations. Hence, a single

fluorescent signal (either mNG or mScarlet) indicates asymmetric HDR events with precise editing at only one end - the 3' or 5' end, respectively. It should be noted that this reporter assay may underestimate the asymmetric HDR with 3' HDR, as frameshifts at the 5' junction prevent mScarlet translation.

To establish the HDR reporter cell line, we utilized the CRISPR/Cpf1 system to insert the reporter cassette into the genome, targeting the C-terminus of HNRNPA1. To use as a readout marker of successful genomic integration, the upstream region of the HDR reporter donor template contains a sequence encoding SNAP-tag, which enables the C-terminus of HNRNPA1 to be endogenously SNAP-tagged (Fig. 4a) Furthermore, in order to exclude mixed signals arising from distinct repair events on the two alleles, we aimed to integrate the HDR reporter cassette into only one allele of the *HNRNPA1* locus. To achieve this, we introduced cells with the SNAP-tag-containing HDR reporter donor together with a HaloTag-tagging donor, both targeting the same locus. Thus, we successfully isolated a cell line with a mono-allelic insertion of the HDR reporter cassette, characterized by the co-expression of HaloTag- and SNAP-tagged HNRNPA1 (Fig. 4b).

Next, we performed knock-in using the donor that contains the complementary halves of mNG and mScarlet in the HDR reporter cell line. Subsequent flow cytometric analysis showed that the major fraction of fluorescence-positive cells exhibited dual fluorescence for both mNG and mScarlet, indicating the occurrence of perfect HDR events (Fig. 4c). Along with that, there was a smaller but considerable portion of single fluorescence-positive cells with only either mNG or mScarlet fluorescence, indicating the presence of asymmetric HDR events. Next, we utilized this reporter system to examine the effects of MMEJ or SSA suppression on the relative occurrence of asymmetric HDR in comparison with perfect HDR. Under NHEJ inhibition, D-I03 treatment decreased the ratio of the single-positive cells to double-positive cells, whereas ART558 had no significant effect (Fig. 4d). This result indicates that SSA suppression specifically reduces asymmetric HDR, consistent with the data of the sequencing analysis using the sorted fluorescence-positive cells (Fig. 3e).

### Discussion

In this study, we comprehensively evaluated the influence of the non-HDR DSB repair pathways on the repair outcomes in CRISPR-mediated endogenous tagging in human cells. Quantitative analysis revealed that NHEJ is the predominant non-HDR repair pathway as shown in previous studies[9,17], while diverse patterns of imprecise repair still frequently occur under NHEJ inhibition. Further inhibition of key factors in the MMEJ and the SSA pathways, POLQ and Rad52 respectively, reduced distinct imprecise repair events and thereby enhanced efficiency of precise knock-in.

In Cpf1-mediated endogenous tagging of HNRNPA1, the suppression of MMEJ using the POLQ inhibitor reduced overall deletion events and increased perfect HDR frequency under NHEJ inhibition, consistent with a previous study[15]. Among the deletions, those occurring in a region flanked by a pair of consecutive T-rich domains (five and six base pairs in length)

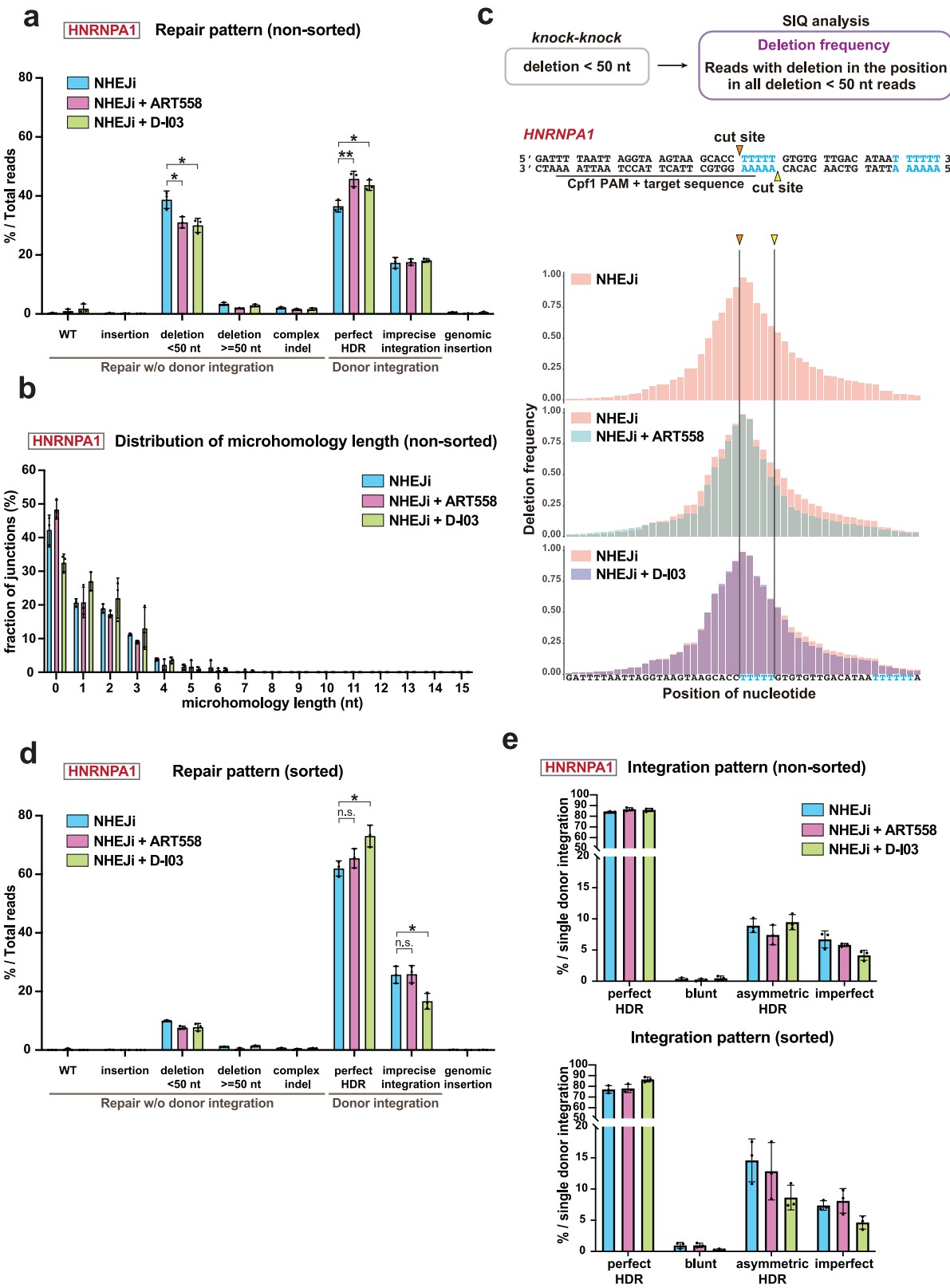

exhibited particularly high susceptibility to POLQ inhibition. This can be attributed to the availability of multiple combinations for microhomology annealing using these T-rich sequences. Interestingly, Rad52 inhibition led to a more selective reduction in the deletions within the same region. One of the T-rich sequence pair is located in the 5' ssDNA overhang region generated by Cpf1 cleavage. Rad52 is known to bind to ssDNA and facilitate the

annealing of complementary sequences during the SSA repair[19]. However, the homology length between the T-rich sequences, at most five nucleotides, is considered to be insufficient for typical SSA-mediated repair[11,38]. Rather, the binding and stabilization of ssDNA overhang by Rad52 may specifically enhance accessibility of this region to POLQ and thereby promote MMEJ. Thus, we propose a potential cooperation between the POLQ-mediated

**Fig. 3 | Inhibition of the SSA pathway contributes to precise DSB repair under NHEJ inhibition. a** Distribution of repair patterns in Cpf1-mediated mNG tagging of HNRNPA1 in cells treated with the indicated inhibitors for 24 h after electroporation. Long-read amplicon sequencing and subsequent *knock-knock* analysis were performed following the methodology depicted in Fig. 1e. 115,790-230,164 sequencing reads were analyzed for each sample. **b** Distribution of microhomology length at genome-donor junctions in Cpf1-mediated mNG tagging of HNRNPA1 in **a**. Reads categorized as the imprecise integrations, where both sides of the donor were trimmed or one side was integrated via HDR while the other was trimmed, were analyzed. **c** Positional distribution of deletions within reads categorized as "deletions of less than 50 nt" by *knock-knock* in **a**. The percentage of reads with a deletion at each nucleotide was calculated and visualized using SIQ program and SIQPlotteR. A

total of 133,684-186,231 reads from three biological replicates for each condition were analyzed and plotted collectively. **d** Distribution of repair patterns in mNG tagging of HNRNPA1 within fluorescence-positive cells. Following the same knock-in and incubation procedure as in **a**, mNG-positive cells were sorted using flow cytometry and subjected to sequencing and *knock-knock* analysis. 147,266-226,842 reads were analyzed for each sample. **e** Distribution of donor integration patterns from *knock-knock* analysis of non-sorted (**a**) and sorted cells (**c**). For each category, the percentage within single donor DNA integration events (perfect HDR, blunt integrations, asymmetric HDR, and imperfect integrations) was calculated. Data from three biological replicates are presented as mean ± S.D. and *P*-value was calculated by a Tukey–Kramer test in this figure. *$P < 0.05$, **$P < 0.01$, n.s. Not significant.

MMEJ pathway and the Rad52-mediated SSA pathway to promote the repair process. This notion is further supported by synergistic increase in knock-in efficiency observed at the *HNRNPA1* locus under simultaneous inhibition of POLQ and Rad52.

In contrast to *HNRNPA1* and *HNRNPA3*, the inhibition of the SSA pathway did not lead to an increase in knock-in efficiency at the other examined gene loci, *TOMM20* and *RAB11A*. The lower sensitivity to SSA inhibition can be attributed to the absence of ssDNA overhangs containing abundant microhomology to nearby regions, which could otherwise promote nucleotide deletions. For instance, at the *RAB11A* locus, DNA cleavage by Cas9 generates DSBs with blunt ends, which lack ssDNA overhangs[39]. At the *TOMM20* locus, Cpf1 produces DSBs with ssDNA overhangs that exhibit significantly less microhomology with neighboring regions compared to those at the *HNRNPA1* and *HNRNPA3* loci (Figure S3e). Thus, the observed correlation between sensitivity to SSA inhibition and the presence of microhomologous ssDNA overhangs supports our hypothesis that Rad52 facilitates nucleotide deletions via microhomology annealing with ssDNA 5' overhangs.

Regarding repair patterns associated with knock-in donor insertion, our observation indicates that the POLQ-mediated MMEJ pathway has minimal contribution to imprecise donor integration (Fig. 3). In contrast, in the case of random integration of exogenous dsDNA, previous studies have shown that POLQ substantially contributes to genomic insertion of dsDNA[35,36]. This difference in contribution may be explained by the presence of HA sequences (90 bp) in the knock-in donors. Notably, POLQ shows preference for short microhomology, as it has been reported to be crucial in repair events involving short homologous sequences (6 nt) but not for those with longer ones ( ≥ 18 nt) flanking the DSB site[38]. Therefore, the presence of long homology regions between the target site and HAs of the donor DNA possibly impedes POLQ-mediated donor integration.

In previous studies, the SSA pathway was suggested to potentially facilitate precise donor integration, by promoting the annealing of the HA sequences in donor DNA to the target genomic sequence[40–42], although no prior experimental evidence supports this hypothesis to our knowledge. The observations in this study show that inhibiting the SSA pathway did not lead to reduction in perfect HDR, but instead reduced asymmetric HDR. This suggests that the SSA pathway likely promotes precise donor insertion at one end, but if the insertion at the other end is carried out imprecisely by alternative repair pathways, it can lead to imprecise donor integration in an asymmetric manner. Thus, inhibiting SSA could suppress imprecise integration, presenting a valuable approach to enhance precise knock-in.

When practically performing CRISPR/Cas-mediated knock-in, how the induced DSBs are repaired can be influenced by the genomic environment of the specific target loci and the experimental system being utilized. Indeed, our study provides evidence that the distribution of repair outcomes, as well as the impact of inhibiting DSB repair pathways, can vary depending on the target gene locus and the type of Cas nuclease used. Moreover, in endogenous tagging with fluorescent proteins, we show that employing fluorescence sorting to enrich knock-in-positive cells influences their repair patterns, consequently altering the impact of repair pathway inhibitions on final knock-in outcomes. Considering these contextual

factors, we propose that the efficiency of precise CRISPR/Cas-mediated knock-in can be effectively enhanced by appropriately employing tailored combinations of DSB repair pathway inhibition approaches.

## Material and methods

### Cell culture
RPE1 cells obtained from the American Type Culture Collection (ATCC) were maintained in Dulbecco's Modified Eagle Medium/Nutrient Mixture F-12 (DMEM/F-12, Nacalai Tesque) with 10% FBS, 100 U/mL penicillin, and 100 μg/mL streptomycin, respectively. Cells were cultured at 37 °C in a humidified 5% $CO_2$ incubator.

### Chemicals
Alt-R HDR enhancer V2 from Integrated DNA Technologies (IDT) was used as NHEJi at 1 μM and stored at −20 °C according to the manufacturer's guidelines. ART558 (MedChemExpress) and D-I03 (MedChemExpress) were dissolved as 10 mM stock in DMSO, stored at −80 °C, and used with a final concentration of 10 μM in medium.

### DNA donor and guide RNA preparation
DNA donors and guide RNAs (sgRNA for Cas9 and crRNA for Cpf1) were prepared according to previously published methods[22,43]. Donor DNAs for C-terminal tagging, containing the 5xGA linker-mNG/mScarlet sequence, and for N-terminal tagging, containing the mNG/mScarlet-5xGA linker sequence, were amplified by PCR from plasmids encoding respective sequences. Two primers were used, each containing a 90-base left or right homology arm (HA) sequence. We used Q5 High-Fidelity 2X Master Mix (New England Biolabs) for PCR. DpnI and Exonuclease I were used for digestion of residual template plasmids and primers, respectively. The DNA donors were then column-purified using the NucleoSpin Gel and PCR Clean-up kit (Macherey-Nagel) and stored at −20 °C or directly used for electroporation. As for mNG tagging of CLTA, donor DNA was prepared with additional 2nd-round PCR using a gel-purified 1st PCR product as a template to minimize non-specific products. All primer sequences used in this study are listed in Supplementary Table 1. A guide RNA (sgRNA for Cas9 and crRNA for Cpf1) was transcribed in vitro from PCR-generated DNA templates. A template DNA containing T7 promoter and sgRNA sequence was amplified by PCR. After treatment of DNase I (Takara Bio), the synthesized guide RNA was purified using the RNA Clean & Concentrator Kit (Zymo Research). All target site sequences of guide RNAs used in this study are listed in Supplementary Table 2.

### Gene knock-in using the CRISPR/Cpf1 and CRISPR/Cas9 system
Endogenous gene tagging using the CRISPR/Cpf1 system was performed with the electroporation of Cpf1-RNP and DNA donors using the Neon Transfection System (Thermo Fisher Scientific) according to the manufacturer's protocol. A.s. Cas12a Ultra (1 μM) (Cpf1) from Integrated DNA Technologies (IDT) and crRNA (1 μM) were pre-incubated in resuspension buffer R (Thermo Fisher Scientific) at room temperature and mixed with cells ($0.125 \times 10^5$/μL), Cpf1 electroporation enhancer (1.8 μM, IDT), and

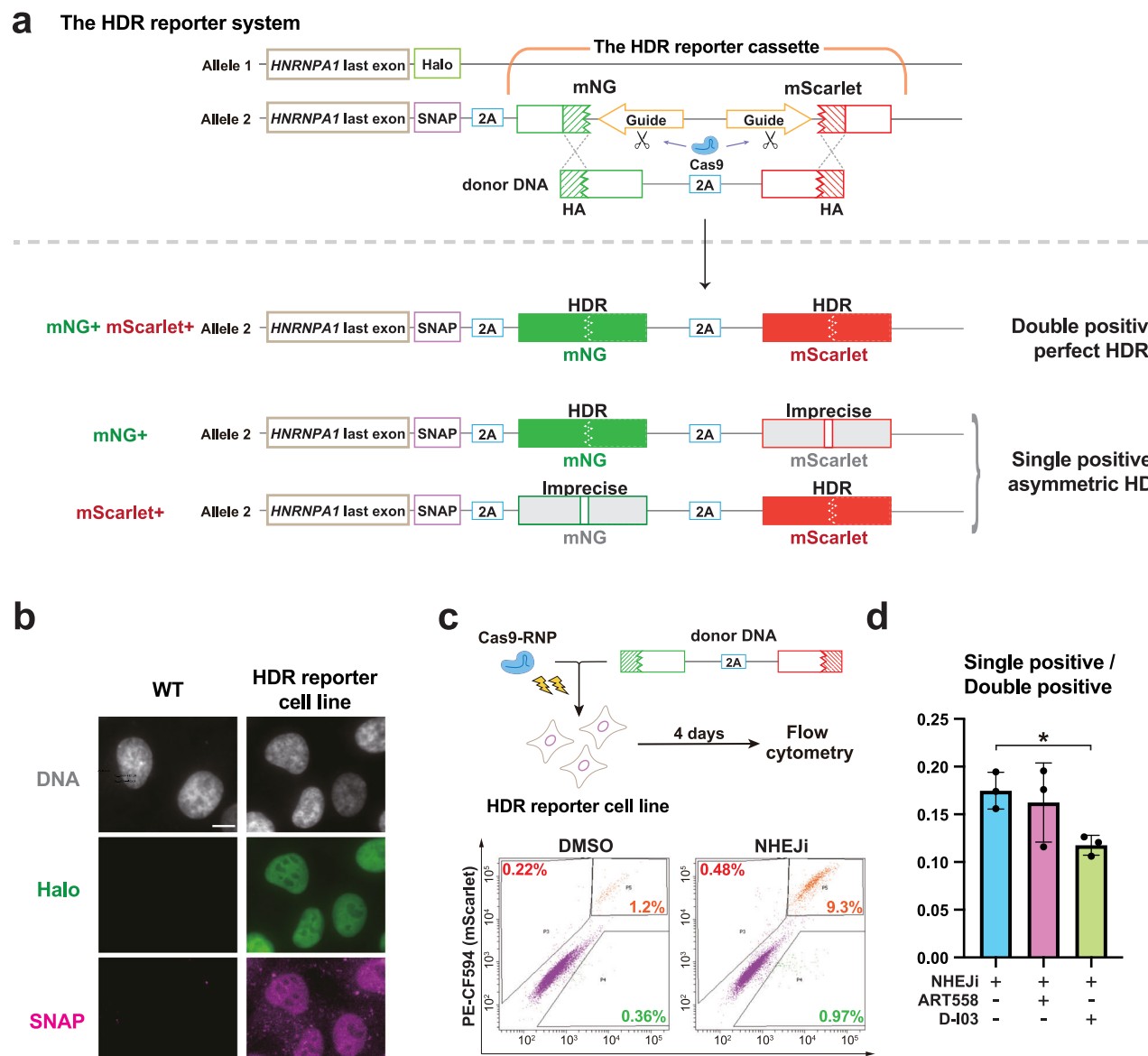

**Fig. 4 | SSA suppression reduces asymmetric HDR repair in the HDR reporter assay. a** Overview of the HDR reporter system. The HDR reporter cassette is monoallelically inserted at the immediate upstream of the stop codon of *HNRNPA1* in RPE1 cells. The seamless, perfect HDR repair of the Cas9-induced DSBs in the reporter cassette completes the two split fluorescent protein sequences, resulting in their simultaneous fluorescence emission (Double positive). The presence of either mNG or mScarlet fluorescence alone (Single positive) shows asymmetric HDR repair with precise editing only at the 3' or 5' end, respectively. The sequence of the donor DNA is shown in Supplementary Table 3. **b** Representative images of the HDR reporter cell line exhibiting both SNAP-tag and HaloTag ligands signals in the localization corresponding to the nuclear protein HNRNPA1. The

parental wild type (WT) cells were analyzed as a control. Scale bar: 10 μm. **c** Time course of the HDR reporter assay and a representative plot obtained through flow cytometric analysis. After electroporation, the cells were treated with DMSO or NHEJi for 24 h. The plot displays the percentages of cells exhibiting only mNG signal (green), only mScarlet signal (magenta), or both signals (orange). **d** The ratio of the single-positive cells to double-positive cells in the HDR reporter assay. Cells were treated with the indicated inhibitors for 24 h after electroporation. Data from three biological replicates are presented as mean ± S.D. and a two-tailed, unpaired Student's t test was used to obtain the *P*-value. *$P < 0.05$.

single or dual DNA donors (33 nM in total, i.e., 16.5 nM for each DNA donor in dual-color tagging). Electroporation was conducted using a 10 μL Neon tip at a voltage of 1300 V with two 20 ms pulses. The transfected cells were seeded into a 24-well plate with medium containing inhibitors. 24 h after electroporation, the inhibitors were removed by replacing the culture medium three times. CRISPR/Cas9-mediated knock-in was performed similarly to the Cpf1-RNP condition described above, with a modification in the electroporation solution. Briefly, HiFi Cas9 protein (1.55 μM, IDT) and sgRNA (1.84 μM) were pre-incubated in buffer R and mixed with cells, Cas9 electroporation enhancer (1.8 μM, IDT), and DNA donors. Electroporation was conducted at a voltage of 1300 V with two 20 ms pulses.

### Fluorescent protein imaging in CRISPR-mediated mNG tagging of HNRNPA1 and RAB11A

6 days after electroporation, cells cultured on coverslips (Matsunami) were fixed with 4% PFA at room temperature for 15 min. After PBS washing, fixed cells were subjected to permeabilization with PBS containing 0.1% Triton X-100 for 10 min at room temperature. The coverslips were washed with PBS and mounted onto glass slides (Matsunami) using ProLong Gold Antifade Mountant with DAPI (Invitrogen), with the cell side down. The representative images were acquired using Axio Imager.M2 microscope (Carl Zeiss) equipped with a 63× lens objective.

### Quantification of knock-in efficiency by flow cytometry

Flow cytometric analysis was conducted 4 days after electroporation. Cells were harvested with trypsin/EDTA solution and suspended in PBS. The cell suspensions were analyzed using BD FACS Aria III (BD Biosciences), equipped with 355/405/488/561/633 nm lasers to detect cells with mNG or mScarlet signal. Data were collected from more than 5000 gated events.

### siRNA-mediated gene knockdown

The following siRNAs were used: Silencer Select siRNA (Life Technologies) against POLQ no.1 (s21059), POLQ no.2 (s21060), Rad52 no. 1 (s11546), Rad52 no. 2 (s11747), and negative control (4390843). Lipofectamine RNAiMAX (Thermo Fisher Scientific) was used with a final concentration of 20 nM total siRNA (Two siRNAs against the same gene were mixed to a final concentration of 10 nM, respectively) for siRNA transfection according to manufacturer's protocol. Transfected cells were cultured for 24 h before electroporation.

### Amplicon sequencing and analysis by *knock-knock* and SIQ

**Genomic DNA preparation**. For amplicon sequencing analysis conducted without the flow cytometric sorting, after electroporation of Cas-RNP targeting *HNRNPA1*, *TOMM20*, *RAB11A* or *CLTA* loci and a donor DNA, the cells were cultured in media containing inhibitors for 24 h, followed by an additional 3 day culture in fresh media. Genomic DNA was then extracted using NucleoSpin DNA RapidLyse kit (Macherey-Nagel). For amplicon sequencing analysis performed for sorted fluorescence-positive cells, following electroporation of Cpf1-RNP targeting *HNRNPA1* or *CLTC* loci and the donor DNA, the cells were cultured in media with inhibitors for 24 h. After subsequent incubation in fresh media for 3 days, 10,000 mNG positive cells were collected using FACS Aria III, followed by genomic DNA extraction using NucleoSpin Tissue XS kit (Macherey-Nagel).

**Amplicon sequencing**. Amplicon libraries were generated using a 2-step PCR and adapter ligation protocol based on the instructions by Pacific Biosciences (Part Number 101-791-800 Version 02, April 2020) with slight modifications. In the first round of PCR, a region flanking the target site of mNG insertion was amplified from the extracted genomic DNA using KOD One Master Mix (TOYOBO) and primers with universal sequences that provide an annealing site for a barcoded primer. These universal sequences served as annealing sites for barcoded primers. The amplified DNA was purified using AMPure XP beads (Beckman Coulter). Subsequently, the purified DNA underwent a second round of PCR using primers from the Barcoded Universal F/R Primers Plate-96v2 (Pacific Biosciences), followed by purification using AMPure PB beads (Pacific Biosciences). The barcoded amplicons were analyzed using TapeStation (Agilent Technologies) and Qubit Fluorometer (Thermo Fisher Scientific). Finally, all the amplicons were pooled together in equimolar amounts as a single sample. The pooled sequencing libraries were prepared using the SMRTbell Template Prep Kit 3.0 (Pacific Biosciences, CA, USA) and loaded onto the PacBio Sequel II or IIe system with Sequel II Binding Kit 3.2, Sequel II Sequencing Kit 2.0 and Sequencing Primer V3.2 (Pacific Biosciences, CA, USA). Sequencing was performed on three Sequel II 8 M SMRT cells, each with a 30 h movie time. The HiFi reads were generated from the full-pass subreads using the DeepConsensus v1.2 program[44], and then demultiplexed by sample barcodes using the SMRT Link software (v11.0.0.146107, v12.0.0.177059 or v13.1.0.221970; Pacific Biosciences, CA, USA). A total of 8,782,430 barcoded reads with a quality score of >= 40 was selected. Detailed information about the sequencing results is provided in Supplementary Table 3.

**Analysis of knock-in outcomes by knock-knock**. After trimming the universal sequences from the reads, the trimmed reads were subjected to analysis using *knock-knock*, a computational pipeline developed by Canaj et al. (2019), to examine the knock-in outcomes. We used a software package of *knock-knock* available at https://github.com/jeffhussmann/knock-knock[6]. Using the broad classification by *knock-knock*, repair patterns designated as "blunt misintegration," "incomplete HDR," "donor fragment," "complex misintegration," and "concatenated misintegration" were collectively categorized as "imprecise integration" in Fig. 1, Fig. 3, Figure S2 and Figure S4. Based on the detailed classification provided by *knock-knock*, repair patterns that were classified as "5' blunt, 3' blunt" were grouped as "blunt". Repair patterns classified as "5' HDR, 3' blunt", "5' blunt, 3' HDR", "5' HDR, 3' imperfect", and "5' imperfect, 3' HDR" were categorized as "asymmetric HDR". Repair patterns classified as "5' blunt, 3' imperfect", "5' imperfect, 3' blunt", and "5' imperfect, 3' imperfect" were assigned to the "imperfect" category in Fig. 3 and Figure S4. These repair patterns—"blunt", "asymmetric HDR", and "imperfect"—together with "perfect HDR" classified by *knock-knock* were collectively categorized as "single donor integration" in Fig. 3 and Figure S4.

**Analysis of microhomology length distribution**. The frequency distribution of microhomology lengths at genome-donor junctions was analyzed in reads classified as "incomplete HDR" or "donor fragment" by *knock-knock*. For "incomplete HDR" reads, microhomology lengths were evaluated at the genome-donor junction of the non-HDR end, whereas for "donor fragment" reads, genome-donor junctions at both ends of the donor were examined. Each frequency was calculated as a percentage by dividing the number of genome-donor junctions with each microhomology length across all the analyzed reads by the total number of analyzed junctions in the dataset and multiplying by 100.

**Analysis of deletion patterns by SIQ**. Reads classified to deletions of less than 50 nt by *knock-knock* were further analyzed by Sequence Interrogation and Quantification (SIQ) program[37]. SIQ provides a comprehensive analysis of a mutation profile at the target site that can be visualized using SIQPlotteR. We used a software package available at https://github.com/RobinVanSchendel/SIQ.

### Establishment of the HDR reporter

**Preparation of donor DNA for knock-in of the HDR reporter cassette to HNRNPA1**. Donor DNA for knock-in of the HDR reporter cassette was prepared by PCR using a plasmid that had been cloned to include SNAP-tag, 2A peptide, and the HDR reporter cassette in this order flanked by HAs for knock-in at the *HNRNPA1* locus. The HaloTag donor was PCR-amplified from a plasmid encoding the 5xGA-HaloTag sequence using two primers containing left and right HAs for knock-in at the *HNRNPA1* locus. The sequence of the donor DNA containing SNAP-tag, 2A peptide, and the HDR reporter cassette are listed in Supplementary Table 3.

**The design of the guide sequence for Cas9**. The guide sequence candidates were obtained by searching for guide sequences targeting some mouse genes using the CRISPR-Cas9 guide RNA design checker provided by IDT. Subsequently, the frequency of off-target sequences in the human genome was assessed for each guide sequence using the CRISPR-Cas9 guide RNA design checker, and the guide sequence targeting mouse *Adenylate kinase 2* with the minimal off-targets was chosen (Supplementary Table 2).

**Isolation of single cells**. A few weeks after electroporation of Cpf1-RNP targeting the *HNRNPA1* locus and two DNA donors (The HDR reporter cassette donor and HaloTag donor), cells were labeled with HaloTag Oregon Green (Promega) according to the manufacturer's protocol. Halo-positive single cells were isolated into a 96-well plate using BD FACS Aria III Cell Sorter. After 17 days culture, the cell clones were seeded into two 96-well plates, one for expansion and the other for visualization. On the following day, cells in the latter plate were labeled with HaloTag Oregon Green and SNAP-Cell 647-SiR (New England

Biolabs) simultaneously according to the manufacturers' protocols. SNAP/Halo double-positive clones were selected after observation using CQ1 Benchtop High-Content Analysis System (Yokogawa Electric Corp). After cell expansion, the SNAP/Halo-positive clone cultured on coverslips (Matsunami) were treated with HaloTag Oregon Green and SNAP-Cell TMR-Star (New England Biolabs) according to manufacturers' protocols and fixed with 4% PFA in PBS at room temperature for 15 min. The coverslips with fixed cells were washed with PBS and mounted onto glass slides using ProLong Gold Antifade Mountant with DAPI, with the cell side down. The representative images of the HDR reporter cell line were acquired using Axio Imager.M2 microscope equipped with a 63× lens objective.

## Statistics and Reproducibility
Statistical comparison between the data from different groups was performed in PRISM v.9 software (GraphPad) using either a Tukey–Kramer test or a two-tailed, unpaired Student's t test as indicated in the figure legends. *P*-values < 0.05 were considered statistically significant. All data shown are mean ± S.D. Sample sizes are indicated in the figure legends.

## Reporting summary
Further information on research design is available in the Nature Portfolio Reporting Summary linked to this article.

## Data availability
All data supporting the findings of this study are available from the corresponding authors on reasonable request. The source data behind the graphs in the paper can be found in Supplementary Data 1.

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

## Acknowledgements

We thank Miho Kiyooka and Wei Chen at National Institute of Genetics for supporting PacBio sequencing, Dr. Yusuke Kishi at Institute for Quantitative Biosciences at the University of Tokyo for supporting quality control of PacBio library preparation, and the Kitagawa lab members for technical supports and helpful discussions. This work was supported by JSPS KAKENHI grants (Grant numbers: 18K06246, 19H05651, 20K15987, 20K22701, 21H02623, 22H02629, 22K19305, 22K19370, 22K20624, 23K14176) from the Ministry of Education, Science, Sports and Culture of Japan, the PRESTO program (JPMJPR21EC) of the Japan Science and Technology Agency, Takeda Science Foundation, The Uehara Memorial Foundation, The Research Foundation for Pharmaceutical Sciences, Koyanagi Foundation, The Kanae Foundation for the Promotion of Medical Science, Kato Memorial Bioscience Foundation, Tokyo Foundation for Pharmaceutical Sciences, The Naito Foundation, Mochida Memorial Foundation for Medical and Pharmaceutical Research, Princess Takamatsu Cancer Research Fund, The Inamori Foundation, Astellas Foundation for research on metabolic disorders, The Kishimoto Fund Research Grant from the Senri Life Science Foundation, and The Sumitomo Foundation.

## Author contributions

S.H. conceived and designed the study. C.T. designed and performed most of the experiments. A.M. and M.G. optimized the genome editing conditions. A.M. and C.T. designed the HDR reporter system. S.O. validated the completion of the HDR reporter cell clone. C.T., A.M. and K.K.I. analyzed the PacBio data with *knock-knock*. M.F., S.Y. and T.C. provided suggestions. A.T. performed PacBio sequencing. C.T., S.H., A.M. and D.K. analyzed the data. C.T., A.M., S.H. and D.K. wrote the manuscript. All authors contributed to discussions and manuscript preparation.

## Competing interests

The authors declare no competing interests.
