## [Transparent Peer Review file · Communications Biology]

Comparative analysis of multiple DNA double-strand break repair pathways in CRISPR-mediated endogenous tagging

Corresponding Author: Dr Shoji Hata

Version 0:

Reviewer comments:

Reviewer #1

(Remarks to the Author)

The manuscript presents valuable insights into the interplay between various DNA double-strand break (DSB) repair pathways in the context of CRISPR-mediated gene knock-in, particularly emphasizing the roles of microhomology-mediated end joining (MMEJ) and single-strand annealing (SSA) in improving knock-in accuracy. The story was interesting. However, I have several points that warrant further clarification and consideration:

- 1, The manuscript mentions a 24-hour treatment period with specific inhibitors. It would be beneficial to discuss whether this duration is adequate to maintain cellular viability and ensure effective inhibition of the targeted pathways. Additionally, how do the authors verify that the inhibitors maintain their efficacy throughout this period?
- 2, The addition of SSA, NHEJ, or MMEJ inhibitors following Cas9-induced DSBs raises concerns about potential effects on cell survival. It would be important to address whether the timing of inhibitor introduction affects cellular health and the overall experimental outcomes.
- 3, There seems to be a discrepancy in the results presented. While the reporter assays suggest that inhibiting either the MMEJ or SSA pathway does not significantly affect knock-in efficiency, the results on line 16 indicate that MMEJ inhibition enhances perfect HDR frequency at the target loci. The authors should provide a more detailed analysis of knock-in efficiency and clearly define the specific types of reporter knock-in being assessed.
- 4, The conclusion on line 20 regarding the impact of SSA pathway suppression on knock-in efficiency being locus-dependent requires further substantiation. It would be beneficial to include additional analyses across various loci to strengthen this claim.
- 5, The manuscript should address whether the use of different inhibitors affects the editing efficiency (Cleavage activity) of Cas9 and Cpf1 at the target sites. This information is crucial for understanding the overall impact of the inhibitors on the CRISPR editing process.
- 6, The assertion that MMEJ and SSA non-HDR pathways influence gene knock-in efficiency upon NHEJ inhibition is intriguing. However, the observation that individual inhibition of MMEJ or SSA does not appear to replicate this phenomenon needs further exploration. The authors should clarify this point and discuss the underlying mechanisms that may contribute to these observations.

Reviewer #2

(Remarks to the Author)

The authors present a detailed analysis of Cas9-induced double-strand break repair outcomes, with a particular focus on HDR events. By tuning the NHEJ, MMEJ, and SSA pathways, they demonstrate that, in addition to NHEJ inhibition, further suppression of MMEJ or SSA can enhance HDR. Notably, the combination of NHEJ and SSA inhibition significantly increases precise HDR while reducing imprecise events such as asymmetric HDR. Overall, the study is well-designed, and the observed HDR repair patterns upon modulation of different DNA repair pathways are intriguing. However, several issues need to be addressed before the manuscript can be considered for publication.

Major:

1. page 7, line 27. The design of the bi-allelic knock-in experiments is problematic. Specifically, double-positive cells represent bi-allelic knock-ins, but single fluorescence-positive cells could also correspond to bi-allelic knock-ins of the same fluorescent protein. As such, the reported double-positive only represents an uncertain fraction of the total bi-allelic knock-in

events. This limitation undermines the claim that "dual-color tagging system is expected to provide a much more sensitive method for evaluating knock-in likelihood". Instead, the total fluorescence-positive population provides a more comprehensive measure of complete fluorescent tag knock-in events, enabling an more unbiased evaluation of HDR efficiency.

2. The interpretation of Figure 3C is incorrect. The figure shows deletion positions (the first nucleotide of each deletion) rather than the full sequences of the deleted regions. Consequently, the data cannot reveal the end points of the deletions. The description of these data on Page 9, Lines 28–33; Page 10, Lines 11–12; and Page 13, Lines 12–15 needs to be revised accordingly. Any conclusions related to MMEJ pathway involvement based on this figure should be reconsidered.

Minor:

1. page 5, line 26, Fig. 1b and S1a. There is no quantitative data supporting the statement that NHEJ inhibition "markedly increases the cell population." Quantitative evidence should be provided or the claim revised.

2. page 7, line 10. Cpf1 generates sticky ends, whereas SpCas9 creates blunt ends. In NHEJ repair, sticky ends are more likely to be digested in the end processing step and result in small deletions. Therefore, the high frequency of deletions <50 nt observed with Cpf1 may be due to NHEJ end processing rather than distinct repair mechanisms triggered by the two nucleases. The authors should analyze the composition of deletions <50 nt to determine whether they primarily result from Cpf1 sticky ends before concluding that different nucleases trigger distinct repair mechanisms.

3. page 9, line 3-4. A recent study (PMID: 38685010) demonstrated that MMEJ inhibition enhances HDR. The authors' data, showing that MMEJ affects perfect HDR significantly, align with these findings. Thus, the statements that their results "challenges previous reports implicating MMEJ in the donor dependent repair process" and "To further validate that MMEJ does not influence donor DNA integration" appear inconsistent. The authors should clarify or reconcile their interpretation with prior studies.

4. page 9, line 16. For Cpf1, sticky ends may lose a few bases during NHEJ repair. For those prevalent sticky base deletions ("TTTTT" between two cut sites), the algorithm will report deletion position in the upstream cut site, this could result in an apparent peak at that location. The authors should verify whether this is the case. If true, the observed "asymmetric deletion pattern" for Cpf1 is not solid, and the related interpretations should be revised or removed.

5. Page 14, line 21. The claim that "This suggests that the SSA pathway is likely to facilitate precise donor insertion at one end, but the subsequent insertion of the other end is not precisely performed, leading to imprecise donor integration in an asymmetric manner" requires stronger evidence. The current data are insufficient to support this conclusion and should be toned down unless additional proof is provided.

6. The title of Figure 3 should be revised to reflect the actual findings. A more accurate title would be: "Inhibition of the SSA pathway contributes to precise DSB repair...". The data do not support the reverse conclusion.

Reviewer #3

(Remarks to the Author)

The manuscript titled "Comparable analysis of multiple DNA double-strand break repair pathways in CRISPR-mediated endogenous tagging" by Tei et al. presents a comprehensive analysis of HDR outcomes by inhibiting three dominant DSB repair pathways. In addition to the primary effector pathway-NHEJ, they revealed that further inhibiting MMEJ or SSA can improve the HDR efficiency via distinct mechanisms. While this study offers valuable insights into the roles of different DNA repair pathways in response to DSBs induced by CRISPR-Cas nucleases, the following points should be addressed.

Major comment:

1. The NHEJ inhibition data looks solid. However, the data from other two pathways, MMEJ and SSA, are less convincing. Most supporting data were derived from a single gene, HNRNPA1, while the TOMM20 data presented in Fig. S3 shows inconsistencies. For example, HDR efficiencies were not improved with MMEJ or SSA inhibition, and no notable differences were observed in other repair patterns in Fig. S3a. This raises concerns that the findings may be locus-specific rather than universally applicable.

Minor comments:

1. The authors should address why no differences were observed in the microhomology length distribution upon MMEJ inhibition in Fig. 3b and Fig. S3b. What about the frequency of total events of microhomology ≥ 2 bp?

2. The Fig. S3d is missed in the figure legend and the text.

Version 1:

Reviewer comments:

Reviewer #1

(Remarks to the Author)

The author's response has effectively addressed all of my concerns. I agree to proceed with publication.

Reviewer #2

(Remarks to the Author)

The authors have addressed all of my concerns. The revised manuscript is improved in both clarity and scientific rigor. I have no further questions and support its publication.

Reviewer #3

(Remarks to the Author)

The authors have addressed all my concerns well. The revised manuscript shows clear improvement. I have no further questions.

Point-by-point response

Manuscript ID --- COMMSBIO-24-8518

We would like to thank all three reviewers for their very positive and constructive comments (typed in brown). We have carefully modified our manuscripts with substantial new data and addressed all of their comments as outlined below (typed in black). We are confident that our manuscript is now ready for publication in *Communications Biology*.

Reviewer #1 (Remarks to the Author):

The manuscript presents valuable insights into the interplay between various DNA double-strand break (DSB) repair pathways in the context of CRISPR-mediated gene knock-in, particularly emphasizing the roles of microhomology-mediated end joining (MMEJ) and single-strand annealing (SSA) in improving knock-in accuracy. The story was interesting. However, I have several points that warrant further clarification and consideration:

We appreciate the reviewer for taking the time to critically read our manuscript for providing insightful feedback. We agree that the main points the reviewer has mentioned are critical for our manuscript. The following are point-by-point responses to the questions.

1, The manuscript mentions a 24-hour treatment period with specific inhibitors. It would be beneficial to discuss whether this duration is adequate to maintain cellular viability and ensure effective inhibition of the targeted pathways. Additionally, how do the authors verify that the inhibitors maintain their efficacy throughout this period?

We thank the reviewer for pointing this out. We consider a 24-hour treatment with the inhibitors to be appropriate. A previous study indicated that HDR occurs within approximately 24 hours after Cas protein delivery (Dodsworth et al., 2020). In addition, NHEJ is activated most rapidly within minutes, followed by MMEJ, whereas SSA and HDR are activated more slowly in DSB repair process (Fu et al., 2021; Oh and Myung, 2022; Scully et al., 2019). Therefore, we believe that a 24-hour duration of inhibitor treatment is reasonable for enhancing the HDR pathway. The concentrations of each inhibitor were determined based on previous studies. The current condition for NHEJi, a concentration of 1 μM for a 24-hour treatment, is recommended by the manufacturer (IDT) (Schubert et al., 2021). The applied condition for ART558, a 10 μM treatment for 24 hours, was used in a prior study,

demonstrating a comparable inhibitory effect on the MMEJ pathway to that achieved by genetic knockout of POLQ (Schimmel et al., 2023). A 10 μ M treatment of D-I03 for 48 hours following DNA cleavage selectively suppresses the SSA pathway in a previous study (Huang et al., 2016). Thus, we applied 10 μ M D-I03 for 24 hours since HDR primarily occurs within 24 hours. Furthermore, our own experiments showed that the current condition for ART558 or D-I03 treatment produced effects in enhancing knock-in efficiency comparable to those achieved with siRNA-mediated POLQ or Rad52 knockdown, respectively (Fig. 2d, f). These data indicate that the 24-hour inhibitor treatment is sufficient for suppressing these DSB repair pathways. The references for these treatment conditions are cited in the results section (page 5, lines 20-25 in the revised manuscript). We have clarified this rationale for determining the inhibitor condition in the result section (page 5, lines 17-19 in the revised manuscript). We also confirmed whether the inhibitor treatment has a severe impact on cell survival in CRISPR-mediated knock-in. After electroporation, cells were treated with each inhibitor for 24 hours and then subjected to an MTT assay on days 1, 2, 3, and 4 post-electroporation to analyze cell survival. As shown in the attached graph (Figure for reviewers 1), although there was a tendency for reduced cell survival in the NHEJi-treated group compared to the DMSO group at day 4 after electroporation, the treatment with each inhibitor under the current condition did not severely reduce cell survival. Therefore, these data indicate that the current treatment conditions of the inhibitors do not cause severe effects on cell survival while maintaining sufficient suppression of the DSB repair pathways.

Figure for reviewers 1

RPE1 cells were electroporated with Cpf1, a guide RNA targeting HNRNPA1, and mNG donor DNA, and seeded into a 96-well plate. After electroporation, cells were cultured for 24 hours in media containing DMSO (control), 1 μ M NHEJi, 10 μ M ART558, or 10 μ M D-I03. Following the 24-hour treatment, the media was replaced with fresh media, and cells were cultured for varying durations depending on the experimental conditions. At the

indicated time point, MTT solution (prepared by mixing medium and 5 mg/mL MTT stock solution at a 10:1 ratio) was added to each well and incubated at 37°C for 2 hours to allow the formation of formazan crystals. The crystals were dissolved using isopropanol, and absorbance was measured at 600 nm using a microplate reader. Cell viability was calculated relative to the DMSO-treated control group at 1 day after electroporation. Data from three biological replicates are represented as mean \pm S.D. and P-values were calculated by a Tukey–Kramer test for the cell viability at day 4 in this figure.

2, The addition of SSA, NHEJ, or MMEJ inhibitors following Cas9-induced DSBs raises concerns about potential effects on cell survival. It would be important to address whether the timing of inhibitor introduction affects cellular health and the overall experimental outcomes.

As mentioned in #1, applying the inhibitors with the current timing and duration – 24 hours following electroporation – effectively inhibits DNA repair pathways without significantly affecting cell survival. Therefore, we consider that the current timing to be appropriate.

3, There seems to be a discrepancy in the results presented. While the reporter assays suggest that inhibiting either the MMEJ or SSA pathway does not significantly affect knock-in efficiency, the results on line 16 indicate that MMEJ inhibition enhances perfect HDR frequency at the target loci. The authors should provide a more detailed analysis of knock-in efficiency and clearly define the specific types of reporter knock-in being assessed.

We thank the reviewer for this comment. As the reviewer pointed out, the increase in knock-in efficiency was not statistically significant under MMEJ inhibition (Fig. 1d). However, the p-value was 0.14 and, regarding the actual values of knock-in efficiency from biological triplicates, the DMSO group are 5.2%, 5.3%, and 5.1%, while the ART558-treated group are consistently higher values of 6.4%, 6.4%, and 6.4%. Thus, MMEJ inhibition is considered to have a tendency to increase knock-in efficiency, while the difference is not statistically significant. On the other hand, perfect HDR was statistically increased under MMEJ inhibition (Fig. 1f). A potential explanation for the lack of a significant increase in the knock-in efficiency quantified by FACS is that perfect HDR is not the major repair pattern among the fluorescent-positive cells. Amplicon sequencing analysis on fluorescent-positive cells revealed that perfect HDR accounted for only about 25% of the repair events, with imprecise integration and indels contributing approximately 45% and 30%, respectively (Figure for reviewers 2), which is consistent with the previous report (Canaj et al., 2019). This new result suggests

that, even at the highest estimate, a figure potentially overestimated due to the diploid cell context, only up to half of the fluorescent-positive cells arise from perfect HDR. The remaining portion, at minimum, results from imprecise donor integration. Therefore, a slight increase in perfect HDR may not be sufficient to contribute to a statistically significant increase in cell population exhibiting fluorescence.

Figure for reviewers 2

Distribution of repair patterns in mNG tagging of HNRNPA1 within fluorescence-positive cells. Cells were electroporated with Cpf1-RNP and mNG donor DNA, and cultured for 4 days. After incubation, mNG-positive cells were sorted by flow cytometry and subjected to sequencing and *knock-knock* analysis. 87,259-132,873s reads were analyzed for each sample.

4, The conclusion on line20 (page 8) regarding the impact of SSA pathway suppression on knock-in efficiency being locus-dependent requires further substantiation. It would be beneficial to include additional analyses across various loci to strengthen this claim.

We thank the reviewer for this important suggestion. To address the reviewer's request, we performed Cpf1-mediated dual-color tagging at a locus other than the SSA-sensitive *HNRNPA1*. The *HNRNPA3* locus exhibits characteristics similar to those of the *HNRNPA1*

locus, where Cpf1-mediated cleavage generates ssDNA overhangs with abundant microhomology to nearby sequences (new data in Fig. S3d). At this locus, suppression of the SSA pathway increased knock-in efficiency under NHEJ inhibition, consistent with the results observed at the *HNRNPA1* locus (new data in Fig. 2g). These results further support that inhibition of the SSA pathway enhances knock-in efficiency at Cpf1-targeted loci in a locus-dependent manner. We have added this new data and revised the results and discussion section accordingly (page 8, lines 15-16 and lines 20-21; page 14, lines 27-29; page 15, line 1 in the revised manuscript).

5, The manuscript should address whether the use of different inhibitors affects the editing efficiency (Cleavage activity) of Cas9 and Cpf1 at the target sites. This information is crucial for understanding the overall impact of the inhibitors on the CRISPR editing process.

We thank the reviewer for this important comment. The most commonly used method to assess the efficiency of Cas protein cleavage activity is the T7E1 assay. This assay detects indel repairs at the target site by T7E1 endonuclease, which recognizes and cleaves mismatched DNA sequences formed by the re-annealing of wild-type DNA and indel-mutated DNA. However, because these indel mutations are generated by the non-HDR DNA repair pathways, including NHEJ, MMEJ, and SSA, inhibition of these pathways leads to alterations in the frequency or the pattern of indel mutations. Therefore, this assay is not suitable for assessing the impact of the repair pathway inhibitors on Cas protein cleavage activity. To analyze the cleavage activity by an alternative method, we focused on the wild-type (WT) allele population, as revealed by amplicon sequencing, since this population includes alleles that were not cleaved by Cas proteins. In the case of Cpf1-mediated mNG tagging of *HNRNPA1*, the WT allele was almost absent (less than 1 %), suggesting that restoration of the Cpf1-mediated cleavage site to WT allele rarely occurs (Fig. 1f). Therefore, we reasoned that quantifying the WT allele population in Cpf1-mediated knock-in experiments could serve as an indicator of Cpf1 cleavage activity. In cells treated with NHEJi, ART558, or D-I03, this WT allele frequency remained unchanged, suggesting that these inhibitors do not affect the cleavage activity of Cpf1 (Fig. 1f). In addition, a previous study on the repair patterns at the *CLTC* locus after Cpf1-mediated mNG tagging showed that NHEJi had no effect on WT allele frequency compared to DMSO control (Takagi et al., 2024). Furthermore, at the *TOMM20* locus, treatment with ART558 or D-I03, in addition to NHEJi, does not significantly alter WT allele frequency (Fig. S3a). These results indicate that Cpf1 cleavage activity is not affected by the inhibitors.

On the other hand, evaluating Cas9 cleavage efficiency from WT allele frequency is

inappropriate because DSBs generated by Cas9 can be accurately repaired through ligation without any base deletions or insertions, restoring WT allele (Guo et al., 2018). This wild-type repair is likely driven by the blunt ligation, one of the two repair mechanisms in the NHEJ pathway (Chang et al., 2017). Indeed, when the end processing pathway, the other NHEJ repair mechanism, was inhibited in Cas9-mediated knock-in, the frequency of WT allele increased, likely reflecting the enhancement of the blunt ligation pathway (Takagi et al., 2024). Thus, due to the presence of a repair mechanism that restores the WT allele, assessing Cas9 cleavage efficiency by WT allele frequency is unsuitable. Therefore, since there is currently no system available to properly evaluate Cas9 cleavage activity in cells, we could not approach the impact of the inhibitors on Cas9 cleavage activity.

6, The assertion that MMEJ and SSA non-HDR pathways influence gene knock-in efficiency upon NHEJ inhibition is intriguing. However, the observation that individual inhibition of MMEJ or SSA does not appear to replicate this phenomenon needs further exploration. The authors should clarify this point and discuss the underlying mechanisms that may contribute to these observations.

We thank the reviewer for this comment. As mentioned in #3, MMEJ inhibition showed a tendency to increase the knock-in efficiency in mNG tagging of HNRNPA1 although the difference is not statistically significant (Fig. 1d). Similarly, for SSA inhibition, the proportion of fluorescent-positive cells from biological triplicates exhibited an increasing trend: 5.2%, 5.3%, and 5.1% in the DMSO group, compared to 5.9%, 6.4%, and 5.4% in the D-103-treated group (Fig. 1d). Therefore, similar to MMEJ inhibition, SSA inhibition also tends to increase knock-in efficiency. The less pronounced effects of inhibiting the MMEJ or SSA pathways alone, compared to those under NHEJ inhibition may be attributed to NHEJ being the dominant repair pathway (Pannunzio et al., 2018). Previous studies indicated that NHEJ suppression increases MMEJ-mediated microhomology repair (Schimmel et al., 2017; Wyatt et al., 2016). Similarly, SSA-mediated repair is also reported to increase with the loss of NHEJ function (Mansour et al., 2008). Therefore, the increased dependency of DSB repair on MMEJ or SSA pathways when NHEJ is deficient may result in the greater increase of knock-in efficiency observed when these minor repair pathways were inhibited in combination with NHEJ inhibition.

Reviewer #2 (Remarks to the Author):

The authors present a detailed analysis of Cas9-induced double-strand break repair outcomes, with a particular focus on HDR events. By tuning the NHEJ, MMEJ, and SSA pathways, they demonstrate that, in addition to NHEJ inhibition, further suppression of MMEJ or SSA can enhance HDR. Notably, the combination of NHEJ and SSA inhibition significantly increases precise HDR while reducing imprecise events such as asymmetric HDR. Overall, the study is well-designed, and the observed HDR repair patterns upon modulation of different DNA repair pathways are intriguing. However, several issues need to be addressed before the manuscript can be considered for publication.

We are sincerely grateful to the reviewer for taking the time to read our manuscript and providing helpful feedback. The following are point-by-point responses to the questions.

Major:

1. page 7, line 27. The design of the bi-allelic knock-in experiments is problematic. Specifically, double-positive cells represent bi-allelic knock-ins, but single fluorescence-positive cells could also correspond to bi-allelic knock-ins of the same fluorescent protein. As such, the reported double-positive only represents an uncertain fraction of the total bi-allelic knock-in events. This limitation undermines the claim that "dual-color tagging system is expected to provide a much more sensitive method for evaluating knock-in likelihood". Instead, the total fluorescence-positive population provides a more comprehensive measure of complete fluorescent tag knock-in events, enabling an more unbiased evaluation of HDR efficiency.

We thank the reviewer for pointing this out and apologize for the inadequacy of our writing in explaining the purpose of the bi-allelic knock-in experiments. In the pre-revised manuscript, we stated that dual-color tagging was performed to detect bi-allelic knock-ins; however, the purpose was actually to detect changes in knock-in likelihood in a highly sensitive manner. As mentioned in previous studies (Naert et al., 2020; Paquet et al., 2016), the probability of bi-allelic editing increases proportionally to the square of the editing efficiency. Indeed, in the dual-color tagging system, the increase in double-positive cells was markedly greater than that in total positive cells—with the latter being approximately equivalent to the increase in the fluorescent-positive cells in the single-color tagging system (Fig. 2c). This difference is illustrated by NHEJ_i treatment, which resulted in more than 20-fold increase over DMSO for double-positive cells compared to a 3-fold increase for total positive cells in the dual-color

tagging system. Thus, we have revised the corresponding result section accordingly (page 7, lines 28-33; page 8 line 1 and lines 5-8 in the revised manuscript).

2. The interpretation of Figure 3C is incorrect. The figure shows deletion positions (the first nucleotide of each deletion) rather than the full sequences of the deleted regions. Consequently, the data cannot reveal the end points of the deletions. The description of these data on Page 9, Lines 28–33; Page 10, Lines 11–12; and Page 13, Lines 12–15 needs to be revised accordingly. Any conclusions related to MMEJ pathway involvement based on this figure should be reconsidered.

We thank the reviewer for this comment. We consider that the SIQPlotterR plots all nucleotides within the deleted regions. To clarify the specification of the SIQPlotterR program, we subjected a single read with a continuous 9-nucleotide deletion (Δ AGCACCTTT) near the cut site of HNRNPA1 to the SIQPlotterR. If, as the reviewer suggested, the first nucleotide of each deletion was plotted, the resulting plot would show a frequency of 1.00 only at the first nucleotide (A) of the 9-nucleotide deletion and a frequency of 0.00 at each downstream deleted nucleotide. However, the actual target plot displayed a frequency of 1.00 at each of the 9 deleted nucleotides (Figure for reviewers 3). This result demonstrates that this plot shows the deletion frequency at each nucleotide position. Related to this, there was a minor mistake in which the nucleotide positions in the target plot were shifted by one base in Figure 3C, but it does not affect the interpretation of the figure. We have corrected this mistake in Figure 3C (No error was found in Figure S3C).

Figure for reviewers 3

Target plot showing the frequency of deletions at each nucleotide position near the cut

site of HNRNPA1. The plot was generated using the SIQ plotter from a single read with the indicated 9-base deletion. Red arrow indicates the first nucleotide (A) of the 9-base deletion.

Minor:

1. page 5, line 26, Fig. 1b and S1a. There is no quantitative data supporting the statement that NHEJ inhibition "markedly increases the cell population." Quantitative evidence should be provided or the claim revised.

We thank the reviewer for this comment. We toned down the sentence to "NHEJi treatment seemed to result in an increase in the cell population with the mNG signal" (page 5, line 25 - line 26 in the revised manuscript).

2. page 7, line 10. Cpf1 generates sticky ends, whereas SpCas9 creates blunt ends. In NHEJ repair, sticky ends are more likely to be digested in the end processing step and result in small deletions. Therefore, the high frequency of deletions <50 nt observed with Cpf1 may be due to NHEJ end processing rather than distinct repair mechanisms triggered by the two nucleases. The authors should analyze the composition of deletions <50 nt to determine whether they primarily result from Cpf1 sticky ends before concluding that different nucleases trigger distinct repair mechanisms.

We thank the reviewer for this comment and apologize for any confusion regarding this claim. As the reviewer pointed out, it is unclear whether the difference in the repair patterns between Cas9- and Cpf1-mediated knock-ins is due to differences in the repair approaches. Therefore, we have revised the manuscript to indicate that the differences in repair patterns arises from the differences in the cleavage ends generated by the Cas proteins (page 7, line 12 in the revised manuscript).

3. page 9, line 3-4. A recent study (PMID: 38685010) demonstrated that MMEJ inhibition enhances HDR. The authors' data, showing that MMEJ affects perfect HDR significantly, align with these findings. Thus, the statements that their results "challenges previous reports implicating MMEJ in the donor dependent repair process" and "To further validate that MMEJ does not influence donor DNA integration" appear inconsistent. The authors should clarify or reconcile their interpretation with prior studies.

We thank the reviewer for this important suggestion and apologize for the inconsistency. To

ensure consistency and accuracy between the two statements, we have revised them to read: "challenges current understanding that MMEJ directly drives imprecise donor integration" and "To further validate that MMEJ is not involved in imprecise donor integration." (page 9, lines 3 - 5 in the revised manuscript).

4. page 9, line 16. For Cpf1, sticky ends may lose a few bases during NHEJ repair. For those prevalent sticky base deletions ("TTTTT" between two cut sites), the algorithm will report deletion position in the upstream cut site, this could result in an apparent peak at that location. The authors should verify whether this is the case. If true, the observed "asymmetric deletion pattern" for Cpf1 is not solid, and the related interpretations should be revised or removed.

We thank for the reviewer this important comment and apologize for any confusions regarding the asymmetric deletion pattern. Our definition of the asymmetric deletion pattern is based on the observation that the upstream of the region flanked by the two cut sites (TTTTT) exhibits a higher frequency of deletions compared to the downstream of the region. Thus, we did not take the overhang sequence (TTTTT) into consideration in the asymmetric deletion pattern. Furthermore, a similar asymmetric deletion pattern was observed at the *TOMM20* locus, where the overhang region (AGCCA) does not contain a T-rich sequence. Therefore, the continuous T sequence was not the cause of the asymmetric deletion pattern. To clearly convey this point, we have revised the results section accordingly (page 9, lines 17-19 in the revised manuscript).

5. Page 14, line 21. The claim that "This suggests that the SSA pathway is likely to facilitate precise donor insertion at one end, but the subsequent insertion of the other end is not precisely performed, leading to imprecise donor integration in an asymmetric manner" requires stronger evidence. The current data are insufficient to support this conclusion and should be toned down unless additional proof is provided.

We thank the reviewer for this insightful suggestion. Your suggestion is appropriate, so we have modified the sentence to: "This suggests that the SSA pathway likely promotes precise donor insertion at one end, but if the insertion at the other end is carried out imprecisely by alternative repair pathways, it can lead to imprecise donor integration in an asymmetric manner." (page 15, lines 21-23 in the revised manuscript).

6. The title of Figure 3 should be revised to reflect the actual findings. A more accurate title would be: "Inhibition of the SSA pathway contributes to precise DSB repair...". The data do

not support the reverse conclusion.

We thank the reviewer for this comment. We have modified the title of Figure 3 as the reviewer requested.

Reviewer #3 (Remarks to the Author):

The manuscript titled “Comparable analysis of multiple DNA double-strand break repair pathways in CRISPR-mediated endogenous tagging” by Tei et al. presents a comprehensive analysis of HDR outcomes by inhibiting three dominant DSB repair pathways. In addition to the primary effector pathway-NHEJ, they revealed that further inhibiting MMEJ or SSA can improve the HDR efficiency via distinct mechanisms. While this study offers valuable insights into the roles of different DNA repair pathways in response to DSBs induced by CRISPR-Cas nucleases, the following points should be addressed.

We are deeply grateful to the reviewer for taking the time to read our manuscript and providing constructive comments. The following are point-by-point responses to the questions.

Major comment:

1. The NHEJ inhibition data looks solid. However, the data from other two pathways, MMEJ and SSA, are less convincing. Most supporting data were derived from a single gene, *HNRNPA1*, while the TOMM20 data presented in Fig. S3 shows inconsistencies. For example, HDR efficiencies were not improved with MMEJ or SSA inhibition, and no notable differences were observed in other repair patterns in Fig. S3a. This raises concerns that the findings may be locus-specific rather than universally applicable.

We thank the reviewer for this important comment. To confirm that the effects of MMEJ or SSA inhibition is not specific to the *HNRNPA1* locus, we performed Cpf1-mediated dual-color tagging at the locus other than *HNRNPA1*. The *HNRNPA3* locus possesses the similar characteristic to the *HNRNPA1* locus, where Cpf1-mediated cleavage generates ssDNA overhangs with abundant microhomology to nearby sequences (new data in Fig. S3d). At this locus, suppression of the MMEJ or SSA pathway increased knock-in efficiency under NHEJ inhibition, consistent with the *HNRNPA1* locus (new data in Fig. 2g). This result

suggests that the enhancement of knock-in efficiency by MMEJ and SSA inhibition is locus specific, attributable to the presence of regions abundant in microhomology to nearby sequences in the ssDNA overhangs generated by Cpf1-mediated cleavage, rather than being specific to the HNRNPA1 locus. We have added the new data and modified the result section accordingly (page 8, lines 15-16 and lines 20-21; page 14, lines 27-29; page 15, line 1 in the revised manuscript).

Minor comments:

1. The authors should address why no differences were observed in the microhomology length distribution upon MMEJ inhibition in Fig. 3b and Fig. S3b. What about the frequency of total events of microhomology ≥ 2 bp?

We thank the reviewer for this valuable suggestion. While MMEJ is known to drive microhomology-mediated repair of DSB ends within the genome, its role in donor-dependent repair at the target site has remained largely unexplored. Figures 3b and S3b present an analysis of microhomology length distributions in imprecise donor-dependent repair, specifically focusing on microhomology at genome-donor junctions. MMEJ inhibition did not alter the distribution of microhomology-mediated repair at genome-donor junctions under NHEJ inhibition. Following the reviewer's suggestion, we also quantified the proportion of total events with microhomology ≥ 2 bp. Notably, the frequency of events with ≥ 2 bp microhomology also showed no significant reduction upon MMEJ pathway inhibition (new data in Fig. S3c). Collectively, these data strongly suggest that MMEJ is not involved in microhomology-mediated repair between the genome and donor DNA. As discussed in the discussion section, POLQ, the core protein of MMEJ, exhibits a preference for short microhomologies, as it has been reported to be critical in repair events involving short homologous sequences (e.g., 6 nt), but not in those with longer flanking sequences (≥ 18 nt) at the DSB site. We hypothesize that the presence of long homology (90 bp) between the genome and donor DNA may suppress POLQ activity. This could explain why the microhomology length distribution remains unaffected despite the inhibition of POLQ-mediated MMEJ. We have added this new data to the supplementary figure and modified the result section accordingly (page 9, lines 8-12 in the revised manuscript).

2. The Fig. S3d is missed in the figure legend and the text.

We thank the reviewer for pointing this out and apologize for the mistake. In the pre-revised

manuscript, we mistakenly referred to Fig. S3d as Fig. S3c. We have corrected the mistake in the results section and the figure legend (page 15, line 1 in the revised manuscript). Please note that due to addition of new figure (Fig. S3d in the revised manuscript), Fig. S3e in the revised manuscript corresponds to Fig. S3d in the original one.

References

- Canaj, H., Hussmann, J. A., Li, H., Beckman, K. A., Goodrich, L., Cho, N. H., Li, Y. J., Santos, D. A., McGeever, A., Stewart, E. M., et al.** (2019). Deep profiling reveals substantial heterogeneity of integration outcomes in CRISPR knock-in experiments. *bioRxiv*.
- Chang, H. H. Y., Pannunzio, N. R., Adachi, N. and Lieber, M. R.** (2017). Non-homologous DNA end joining and alternative pathways to double-strand break repair. *Nat. Rev. Mol. Cell Biol.* **18**, 495–506.
- Dodsworth, B. T., Hatje, K., Meyer, C. A., Flynn, R. and Cowley, S. A.** (2020). Rates of homology directed repair of CRISPR-Cas9 induced double strand breaks are lower in naïve compared to primed human pluripotent stem cells. *Stem Cell Res.* **46**, 101852.
- Fu, Y. W., Dai, X. Y., Wang, W. T., Yang, Z. X., Zhao, J. J., Zhang, J. P., Wen, W., Zhang, F., Oberg, K. C., Zhang, L., et al.** (2021). Dynamics and competition of CRISPR-Cas9 ribonucleoproteins and AAV donor-mediated NHEJ, MMEJ and HDR editing. *Nucleic Acids Res.* **49**, 969–985.
- Guo, T., Feng, Y. L., Xiao, J. J., Liu, Q., Sun, X. N., Xiang, J. F., Kong, N., Liu, S. C., Chen, G. Q., Wang, Y., et al.** (2018). Harnessing accurate non-homologous end joining for efficient precise deletion in CRISPR/Cas9-mediated genome editing. *Genome Biol.* **19**, 1–20.
- Huang, F., Goyal, N., Sullivan, K., Hanamshet, K., Patel, M., Mazina, O. M., Wang, C. X., An, W. F., Spoonamore, J., Metkar, S., et al.** (2016). Targeting BRCA1-and BRCA2-deficient cells with RAD52 small molecule inhibitors. *Nucleic Acids Res.* **44**, 4189–4199.
- Mansour, W. Y., Schumacher, S., Rosskopf, R., Rhein, T., Schmidt-Petersen, F., Gatzemeier, F., Haag, F., Borgmann, K., Willers, H. and Dahm-Daphi, J.** (2008). Hierarchy of nonhomologous end-joining, single-strand annealing and gene conversion at site-directed DNA double-strand breaks. *Nucleic Acids Res.* **36**, 4088–4098.
- Naert, T., Tulkens, D., Edwards, N. A., Carron, M., Shaidani, N. I., Wlizla, M., Boel, A., Demuyne, S., Horb, M. E., Coucke, P., et al.** (2020). Maximizing CRISPR/Cas9 phenotype penetrance applying predictive modeling of editing outcomes in *Xenopus* and zebrafish embryos. *Sci. Rep.* **10**, 1–12.
- Oh, J. M. and Myung, K.** (2022). Crosstalk between different DNA repair pathways for DNA double strand break repairs. *Mutat. Res. - Genet. Toxicol. Environ. Mutagen.* **873**, 503438.
- Pannunzio, N. R., Watanabe, G. and Lieber, M. R.** (2018). Nonhomologous DNA end-joining

for repair of DNA double-strand breaks. *J. Biol. Chem.* **293**, 10512–10523.

- Paquet, D., Kwart, D., Chen, A., Sproul, A., Jacob, S., Teo, S., Olsen, K. M., Gregg, A., Noggle, S. and Tessier-Lavigne, M.** (2016). Efficient introduction of specific homozygous and heterozygous mutations using CRISPR/Cas9. *Nature* **533**, 125–129.
- Schimmel, J., Kool, H., van Schendel, R. and Tijsterman, M.** (2017). Mutational signatures of non-homologous and polymerase theta-mediated end-joining in embryonic stem cells. *EMBO J.* **36**, 3634–3649.
- Schimmel, J., Muñoz-Subirana, N., Kool, H., van Schendel, R., van der Vlies, S., Kamp, J. A., de Vrij, F. M. S., Kushner, S. A., Smith, G. C. M., Boulton, S. J., et al.** (2023). Modulating mutational outcomes and improving precise gene editing at CRISPR-Cas9-induced breaks by chemical inhibition of end-joining pathways. *Cell Rep.* **42**, 112019.
- Schubert, M. S., Thommandru, B., Woodley, J., Turk, R., Yan, S., Kurgan, G., McNeill, M. S. and Rettig, G. R.** (2021). Improved methods and optimized design for CRISPR Cas9 and Cas12a homology-directed repair. *bioRxiv* doi: [10.1101/2021.04.07.438685](https://doi.org/10.1101/2021.04.07.438685)
- Scully, R., Panday, A., Elango, R. and Willis, N. A.** (2019). DNA double-strand break repair-pathway choice in somatic mammalian cells. *Nat. Rev. Mol. Cell Biol.* **20**, 698–714.
- Takagi, R., Hata, S., Tei, C., Mabuchi, A. and Anzai, R.** (2024). Comprehensive analysis of end-modified long dsDNA donors in CRISPR-mediated endogenous tagging. *bioRxiv* doi: [10.1101/2024.06.28.601124](https://doi.org/10.1101/2024.06.28.601124)
- Wyatt, D. W., Feng, W., Conlin, M. P., Yousefzadeh, M. J., Roberts, S. A., Mieczkowski, P., Wood, R. D., Gupta, G. P. and Ramsden, D. A.** (2016). Essential Roles for Polymerase θ -Mediated End Joining in the Repair of Chromosome Breaks. *Mol. Cell* **63**, 662–673.